# Brain-wide projection reconstruction of single functionally defined neurons

Meng Wang[1,2,3,13], Ke Liu[1,2,13], Junxia Pan[1,2,13], Jialin Li[3,13], Pei Sun[1,2,13], Yongsheng Zhang[4,5,13], Longhui Li[3,6], Wenyan Guo[4,5], Qianqian Xin[1,2], Zhikai Zhao[1,2], Yurong Liu[4,5], Zhenqiao Zhou[7], Jing Lyu[7], Ting Zheng[4,5], Yunyun Han[8], Chunqing Zhang[1,2], Xiang Liao[3✉], Shaoqun Zeng[4,5✉], Hongbo Jia[6,7,9,10✉] & Xiaowei Chen[1,2,11,12✉]

Reconstructing axonal projections of single neurons at the whole-brain level is currently a converging goal of the neuroscience community that is fundamental for understanding the logic of information flow in the brain. Thousands of single neurons from different brain regions have recently been morphologically reconstructed, but the corresponding physiological functional features of these reconstructed neurons are unclear. By combining two-photon $Ca^{2+}$ imaging with targeted single-cell plasmid electroporation, we reconstruct the brain-wide morphologies of single neurons that are defined by a sound-evoked response map in the auditory cortices (AUDs) of awake mice. Long-range interhemispheric projections can be reliably labelled via co-injection with an adeno-associated virus, which enables enhanced expression of indicator protein in the targeted neurons. Here we show that this method avoids the randomness and ambiguity of conventional methods of neuronal morphological reconstruction, offering an avenue for developing a precise one-to-one map of neuronal projection patterns and physiological functional features.

[1] Brain Research Center, Third Military Medical University, Chongqing 400038, China. [2] State Key Laboratory of Trauma, Burns, and Combined Injury, Third Military Medical University, Chongqing 400038, China. [3] Center for Neurointelligence, School of Medicine, Chongqing University, Chongqing 400030, China. [4] Britton Chance Center for Biomedical Photonics, Wuhan National Laboratory for Optoelectronics, Huazhong University of Science and Technology, Wuhan, China. [5] MoE Key Laboratory for Biomedical Photonics, School of Engineering Sciences, Huazhong University of Science and Technology, Wuhan, China. [6] Advanced Institute for Brain and Intelligence, Guangxi University, Nanning 530004, China. [7] Brain Research Instrument Innovation Center, Suzhou Institute of Biomedical Engineering and Technology, Chinese Academy of Sciences, Suzhou 215163, China. [8] Department of Neurobiology, School of Basic Medicine and Tongji Medical College, Huazhong University of Science & Technology, Wuhan, China. [9] Institute of Neuroscience and the SyNergy Cluster, Technical University Munich, 80802 Munich, Germany. [10] Combinatorial NeuroImaging Core Facility, Leibniz Institute for Neurobiology, 39118 Magdeburg, Germany. [11] CAS Center for Excellence in Brain Science and Intelligence Technology, Shanghai Institutes for Biological Sciences, Chinese Academy of Sciences, Shanghai 200031, China. [12] Chongqing Institute for Brain and Intelligence, Guangyang Bay Laboratory, Chongqing 400064, China. [13]These authors contributed equally: Meng Wang, Ke Liu, Junxia Pan, Jialin Li, Pei Sun, Yongsheng Zhang. ✉email: xiang.liao@cqu.edu.cn; sqzeng@hust.edu.cn; jiahb@sibet.ac.cn; xiaowei_chen@tmmu.edu.cn

Understanding how neuronal functions are related to morphology is a central goal of modern neuroscience[1–3]. Axonal projections determine how neurons route their physiological functional output information to target brain regions[4,5]. However, even within a small brain region, neurons often exhibit extraordinary functional diversity and are spatially intermingled[6–10]. Thus, there is a great demand for axonal projection reconstructions of neurons that are defined by physiological functional features in a given brain region. In principle, three levels of methods are available for this purpose. The first-level methods enable non-specific labelling of neurons via expression of fluorescent proteins[11–13] for visualization and reconstruction using whole-brain serial sectioning imaging techniques[14–16]. At this level, the neurons to be labelled are usually randomly selected from a defined brain region. The second-level methods label neurons with fluorescent proteins based on specific molecular markers through a Cre-dependent expression procedure to enable the reconstruction of molecularly defined neuronal types[17]. Although highly relevant, molecularly defined cell types do not unambiguously define specific physiological functional features under in vivo conditions. The third-level methods involve both electrophysiological recording and plasmid delivery to express fluorescent proteins in vivo via a patch pipette on the same neurons[18,19]. These methods enable direct matching of single-cell physiological functional features and axonal projections but with the same caveat that chance largely determines which exact neurons are reconstructed. In addition, these methods have not yet yielded a complete reconstruction of neuronal morphology at the whole-brain level.

With the help of either recently developed high-throughput techniques[13,20] or extensive human labour[21], it may someday become possible for neurons with different types of physiological functional features to be sufficiently sampled and reconstructed. However, many neurons with specific response patterns, such as those related to learning, memory, and behaviour, are sparse, constituting only a small percentage of neurons in the relevant brain regions[10,22], which raises concerns that high-throughput techniques may yield highly ambiguous results. Thus, there is a strong but unmet demand for a method that enables precise, targeted labelling of neurons whose physiological functional features have been defined in vivo. Here, we report the 2-SPARSE (two-photon imaging-assisted Single-cell Plasmid electroporation and Adeno-associated virus (AAV) injection for Reconstructing Single nEurons) method that seamlessly combines two-photon $Ca^{2+}$ imaging of cortical neuronal populations in awake mice[10,23] and postmortem whole-brain serial section imaging of dendrites and axonal projections using fluorescence micro-optical sectioning tomography (fMOST)[24]. As an example demonstration of this method, we identified single neurons with specific tone-tuning response profiles[6] in the AUDs of awake mice and electroporated these neurons one by one to label their dendrites and axons. Using retrograde labelling[12] from the distant projection target area as a control, we confirmed that long-range interhemispheric cortico-cortical projections can be reliably labelled by 2-SPARSE. Furthermore, we also validated the labelling efficiency of 2-SPARSE by comparing it in side-by-side experiments with that of other previously reported strategies (e.g., binary AAV expression for labelling sparse neurons). We also demonstrated the extended application of the 2-SPARSE method in other brain regions and deeper layers (e.g., the motor cortex and layer 5). Importantly, 2-SPARSE can be readily implemented in a broad range of laboratories that have been equipped with a standard two-photon microscope and conventional electrophysiological devices.

## Results

**The 2-SPARSE workflow.** Briefly, 2-SPARSE consists of four key steps to achieve brain-wide reconstruction of functionally defined neurons in the mouse brain (see critical milestones in Supplementary Table 1). As an example, we describe the steps used to reconstruct targeted AUD neurons with defined tone-tuning response properties (Fig. 1a and Supplementary Video 1). (1) A bulk population of neurons located in layer 2/3 (L2/3) of the AUD in awake mice was labelled by local injection of a $Ca^{2+}$ fluorescent indicator dye, Cal-520 AM. Sound-evoked $Ca^{2+}$ transients were recorded by two-photon $Ca^{2+}$ imaging, and the frequency response area (FRA) and best frequency (BF) of each neuron were determined[6]. (2) Single neurons with specific tone-tuning properties (as shown in the examples in Figs. 1 and 4) were chosen to be targeted for expression of a Cre-GFP (hSyn-eGFP-P2A-Cre-pA; see "Methods" for details) plasmid by single-cell electroporation[25]. To enhance fluorescent protein expression in long-range projection axons, an AAV that contained a Cre-dependent expression cassette encoding membrane green fluorescent protein (mGFP) (AAV-hSyn-DIO-mGFP) was injected laterally to the site of the electroporated neurons. (3) Thirty days after single-cell electroporation, the animals were perfused, their brains were removed and transferred to the fMOST imaging device, and whole-brain serial sectioning imaging datasets were acquired at submicron voxel resolution. (4) Morphological reconstruction and quantitative analysis were performed to study axonal projection patterns of the imaged neurons. Next, we illustrate how and why we configured the relevant basic techniques and combined them to enable complete reconstruction of brain-wide projections of single functionally defined neurons.

**Online identification of functional features of single neurons in vivo.** In the first step, we used a $Ca^{2+}$ fluorescent indicator dye for two-photon neuronal population imaging in awake mice, recorded sound-evoked $Ca^{2+}$ responses over multiple trials, and then analyzed the FRA and BF for individual neurons (Fig. 1b–d and Supplementary Fig. 1). We first chose two example neurons (Fig. 1b, c) with similar response features according to the definition of BF = 16.3 kHz and targeted them for morphology labelling. In this approach, it is critical to perform acute functional imaging with a synthetic $Ca^{2+}$ dye that is gene free to avoid interference with the subsequent morphological labelling procedure (which relies on genetic approaches).

**Targeted single-cell plasmid electroporation in vivo.** In the second step, we combined two genetic approaches to achieve complete fluorescent labelling of individual neurons. A micropipette (electrical resistance of ~12 MΩ) containing intracellular solution and a Cre-GFP plasmid was advanced towards the soma of each functionally defined target neuron one by one under live two-photon imaging guidance (Fig. 1e). The same micropipette was used to apply electrical current pulses (pulse amplitude: −450 nA, duration: 500 μs per pulse, train of 100 pulses at a frequency of 50 Hz) to perforate the cellular membrane and deliver the plasmid into the cell body. Electroporation was deemed successful when pores on the membrane of the target neuron were first opened by electrical shock. An intracellular $Ca^{2+}$ dye, OGB-1 potassium salt, was added to the micropipette solution to visualize whether the cell body was filled after the electrical pulses (Fig. 1e). We also monitored $Ca^{2+}$ activity to test whether the electroporated neuron remained alive. We compared spontaneous $Ca^{2+}$ transients before (Fig. 2a) and after (Fig. 2b) electroporation and found no significant difference in either signal amplitude ($P = 0.2844$, before: $n = 252$ transients; after: $n = 260$ transients) or frequency ($P = 0.7228$, $n = 78$ trials for both cases, 39 neurons from 16 mice; Fig. 2c, d). The success rate of electroporation reached >90% (36 out of 39 neurons in 16 animals) in consecutive experiments (Supplementary Table 2).

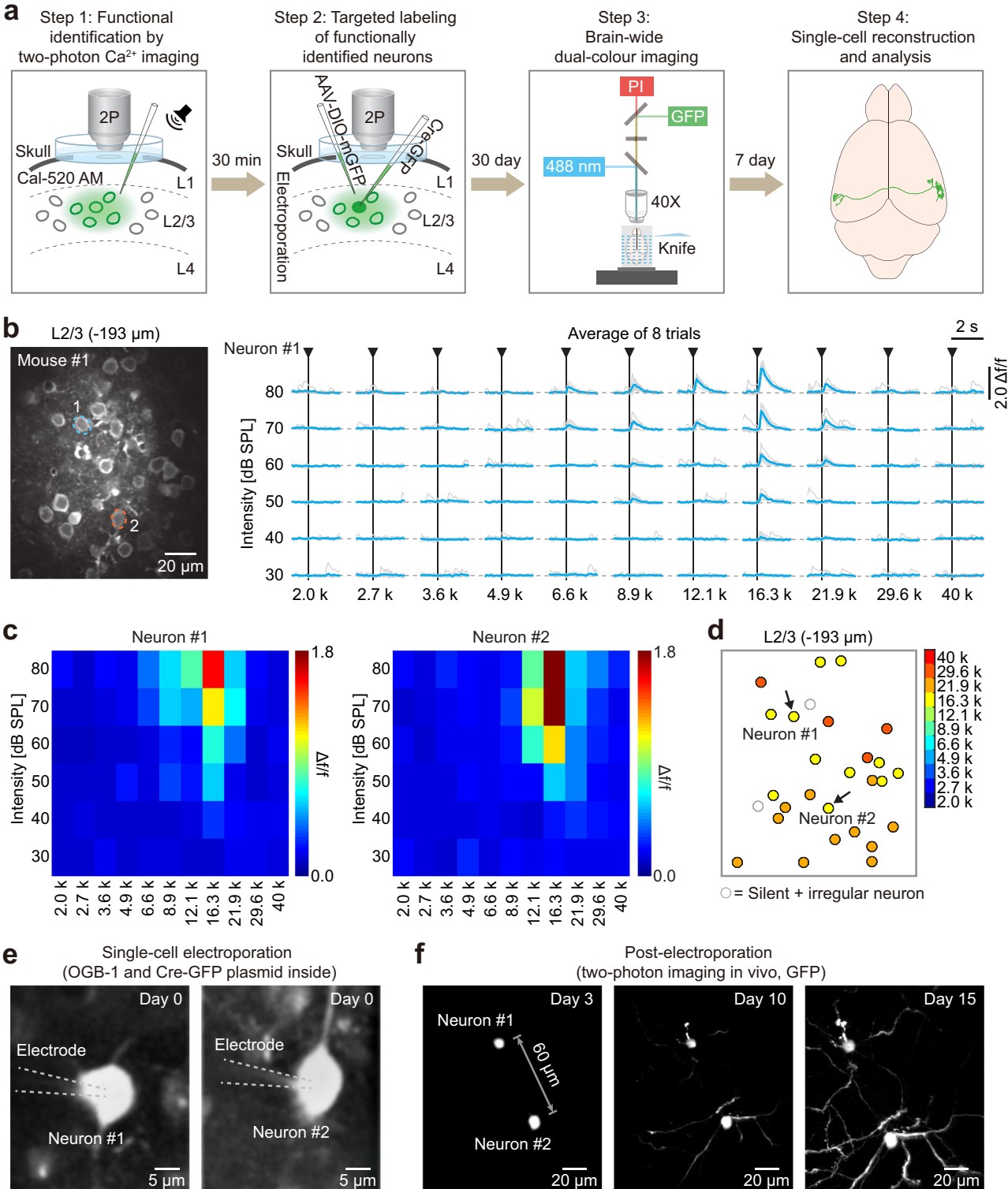

**Fig. 1 Online targeted labelling of single functionally defined neurons. a** Four key steps for data collection and processing. **b** Left: Representative two-photon image in L2/3 of the AUD of an awake mouse. Two neurons were chosen and are marked with dashed coloured lines. Right: Ca$^{2+}$ transients (neuron #1) responding to pure-tone stimulation (average of 8 trials for each of 6 intensities and 11 frequencies). Single-trial (grey) and trial-averaged (blue) Ca$^{2+}$ transients are shown. **c** Colour-coded FRAs of the two neurons marked in (**b**). **d** Best frequency map of all the neurons in the imaging plane shown in (**b**). The colour code is on the right side. **e** Demonstration of single-cell electroporation. Briefly, pipettes containing OGB-1 and Cre-GFP plasmid were advanced towards the somata of neuron #1 and neuron #2. Then, the dye and DNA plasmids were electroporated into the neurons by applying trains of voltage pulses. **f** Monitoring of the fluorescence intensities of the two electroporated neurons by two-photon imaging on different days (day 3, day 10, and day 15) after electroporation.

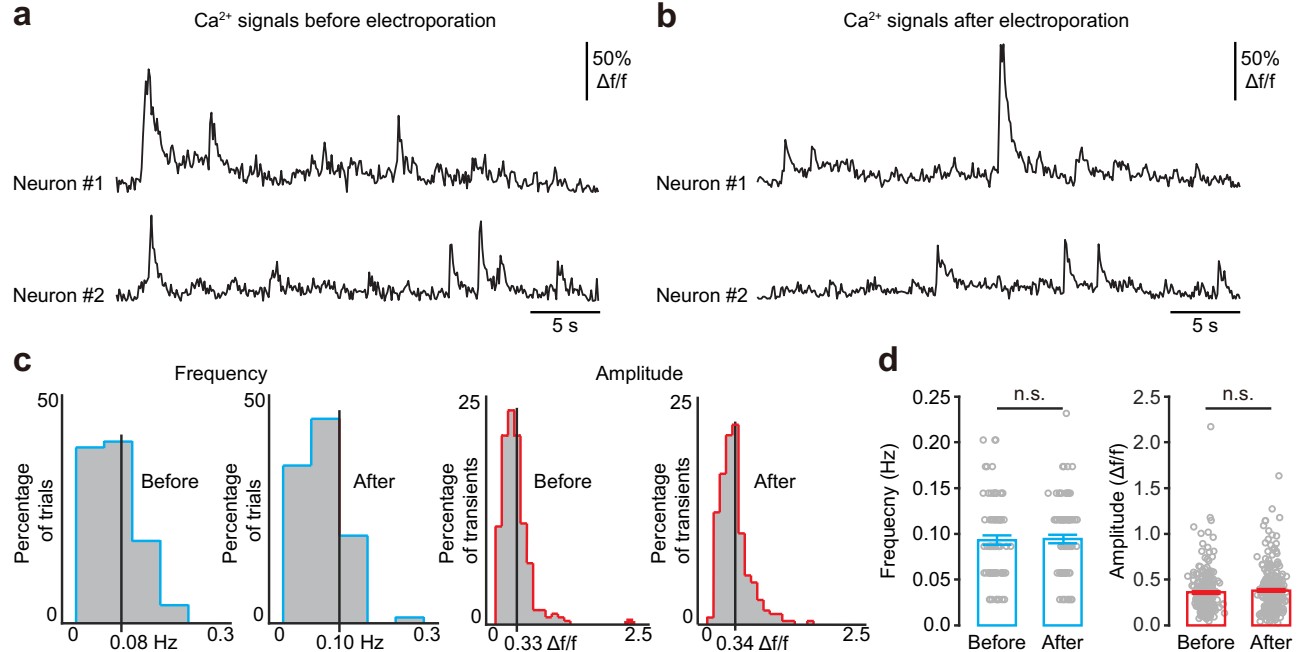

**Fig. 2 Single-cell electroporation did not alter spontaneous neuronal activity. a, b** Spontaneous $Ca^{2+}$ signals of two neurons before (**a**) and one hour after (**b**) electroporation (the same neurons as in Fig. 1b). **c** Distribution of the frequency (left) and amplitude (right) of spontaneous $Ca^{2+}$ transients before and after electroporation. **d** Bar graphs summarizing the frequency ($n = 78$ trials for both cases, 39 neurons, 16 mice; $P = 0.7228$, two-sided Wilcoxon signed-rank test) and amplitude (before: $n = 252$ transients; after: $n = 260$ transients, 39 neurons, 16 mice; $P = 0.2844$, two-sided Wilcoxon rank-sum test, n.s., $P > 0.05$). The data with error bars represent mean ± SEM.

For each animal, after one or several neurons were successfully electroporated, the electroporation pipette was retracted, and another micropipette loaded with AAV solution was inserted through the same track towards the target zone. The AAV contained a Cre-dependent expression cassette (double-floxed inverse open reading frame, DIO) encoding mGFP. The AAV solution was pressure-injected into the extracellular space near the location of the electroporated neurons (see "Methods" for details). The combination of single-cell Cre plasmid electroporation and nearby local AAV injection is a key feature of our method, which utilizes Cre to activate and amplify robust fluorescence expression in targeted neurons without risk of virus spill-over. After in vivo functional identification, targeted electroporation and AAV injection, we sealed the craniotomy with a glass window and monitored the expression level of GFP in the targeted neurons over time (days) through chronic in vivo two-photon imaging (Fig. 1f). Successful labelling was defined by the electroporated neuron remaining clearly visible without signs of damage or death for 30 days after electroporation. The overall success rate of all the steps until full labelling of functionally identified single neurons in vivo reached >80% in consecutive experiments (32 out of 39 neurons in 16 animals, Supplementary Table 2).

The key part of this step is the combination of single-cell plasmid electroporation and nearby local AAV injection to unlock robust fluorescence expression in target neurons. Using this step, we achieve an optimal balance between expression level and cellular toxicity, thus bypassing the challenge of titrating the plasmid concentration. Moreover, the number of labelled neurons per imaging region can be determined by the number of neurons electroporated (Supplementary Fig. 2). Axon crossovers can introduce ambiguity into reconstruction. However, we were able to precisely reconstruct the two neurons with a soma distance of ~100 μm in Fig. 1. Based on this experience, we then used our method to label a maximum of four neurons within a single two-

photon imaging field of view (200 μm × 200 μm), consequently avoiding separability issues with post-mortem morphological reconstruction, which was consistent with a previous study[12].

**Post-mortem whole-brain serial sectioning and imaging.** In the third step, we performed whole-brain serial sectioning and imaging at a resolution of $0.3 \times 0.3 \times 1 \, \mu m^3$ (see "Methods" for details) with the fMOST technique (Supplementary Fig. 3). Thirty days after electroporation, the expression of GFP reached a sufficiently high level for post-mortem fMOST imaging. We chose the fMOST imaging technique because of its high z-resolution, important for resolving axons, which is ensured by mechanical sectioning. We fixed the post-mortem mouse brain samples at the in vivo two-photon imaging setup and then transferred them to the fMOST imaging system, the latter of which also offers globally accessible fMOST imaging services for such a workflow. With our labelling technique from the second step, we were able to clearly identify the soma, dendrites, and distant axons of each targeted neuron in the raw images (Supplementary Fig. 4). The complete morphology of each labelled neuron was reconstructed by manual tracing (Supplementary Fig. 5). In order to sufficiently balance reconstruction accuracy and throughput, each neuron was traced by two experienced annotators, and the consensus result was approved (Supplementary Fig. 6)[26]. Then, all the reconstructed neurons were registered to the template of the Allen Brain Atlas, i.e., the Allen Mouse Common Coordinate Framework (CCF v3), combining the approaches of greyscale-based three-dimensional (3D) registration and dense landmark-based two-dimensional (2D) registration in the local regions (Supplementary Fig. 7).

**Brain-wide reconstruction of individual functionally defined neurons.** In the last step, both the dendrites and axonal projections of labelled neurons were completely traced, reconstructed and analyzed (Fig. 3). In a representative whole-brain dataset consisting of two L2/3 neurons (neurons #1 and #2, the same cells

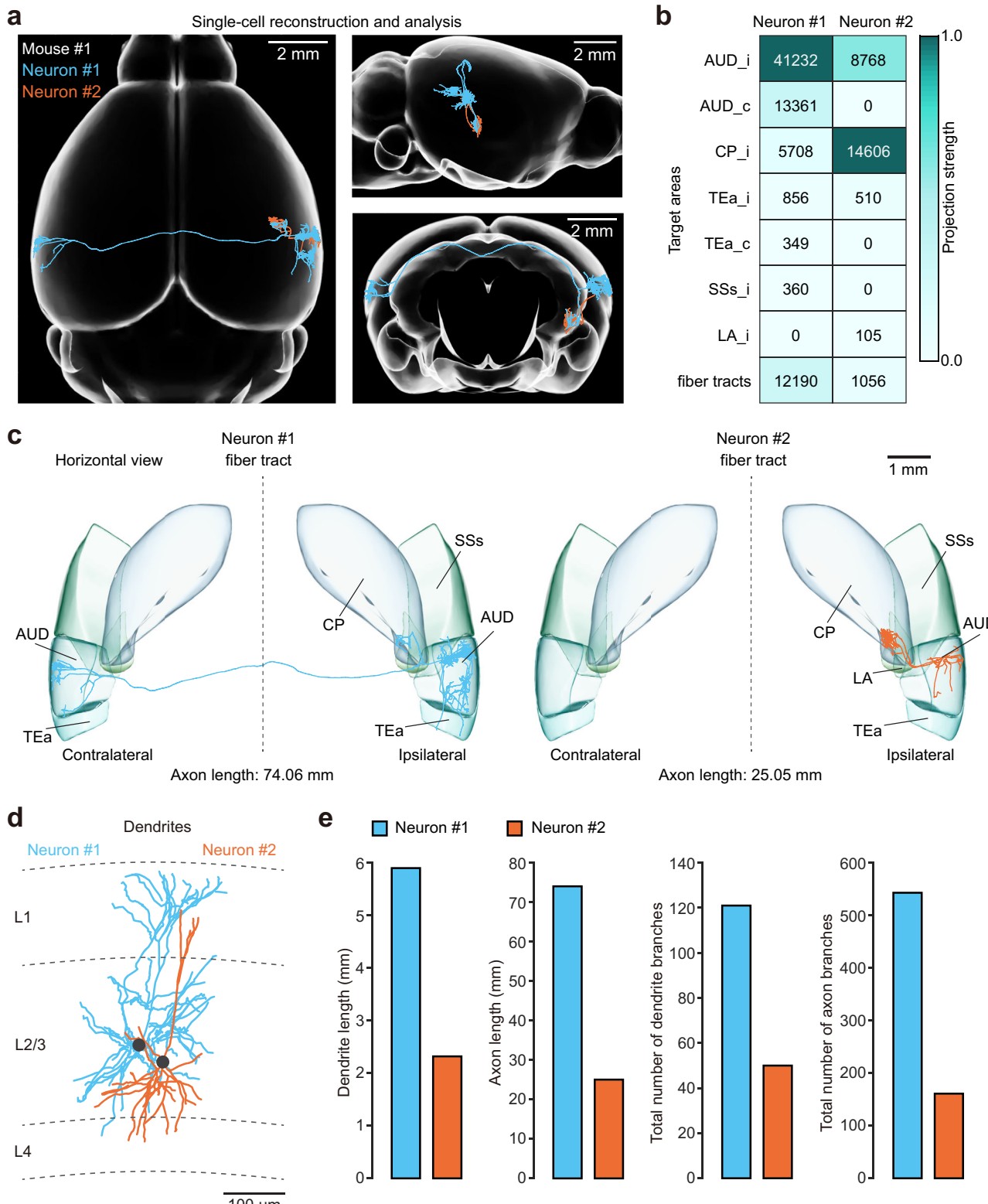

**Fig. 3 Whole-brain reconstruction of individual functionally similar neurons. a** Two reconstructed neurons (neurons #1 and #2) registered to the standard Allen Brain Atlas are shown in horizontal (left), sagittal (upper right) and coronal (lower right) views. **b** Projection strengths of the two neurons. The projection strength of each neuron was calculated as the axon length per target area normalized by the length of the axon receiving the densest innervation. The colour code reflecting the projection strength is shown on the right. **c** The two reconstructed neurons are displayed separately. The target areas are coloured as indicated. **d** The complete dendrites of the two brain-wide reconstructed neurons are displayed in different colours (neuron #1: cyan; neuron #2: orange). The grey dashed lines indicate laminar borders. The two neuronal somata are located in L2/3. **e** Comparisons of dendrite and axon lengths and total numbers of axonal and dendritic branches between neuron #1 and neuron #2. AUD auditory areas; CP caudatoputamen; LA lateral amygdalar nucleus; SSs supplemental somatosensory area; TEa temporal association areas; c contralateral; i ipsilateral.

as in Fig. 1) from the same AUD and with the same tone-responding properties (BF = 16.3 kHz), we found that these two neurons exhibited distinct projection patterns (Fig. 3a) as well as various projection strengths within diverse brain regions (Fig. 3b). For instance, neuron #1 exhibited a bilateral projection pattern, with projections to the ipsilateral auditory cortex (AUD_i), temporal association areas (TEa_i) and caudatoputamen (CP_i), and with a major axon crossing the corpus callosum and finally forming dense axonal terminal fields at the contralateral target areas, mainly the auditory cortex (AUD_c). In contrast, the projections from neuron #2 were mainly distributed in regions including the ipsilateral striatum (CP_i) and auditory areas (AUD_i) (Fig. 3c). We performed a quantitative analysis of both axons and dendrites of these two neurons (Fig. 3c, d) and found that the length and total number of branches of the axons were much larger for neuron #1 than for neuron #2. As axonal length per area in the cerebral cortex has previously been reported to correlate strongly with the number of recipient neurons[27], these results suggested that neuron #1 may have more output recipients than neuron #2. In addition, the length and total number of branches of dendrites were also much larger for neuron #1 than for neuron #2, indicating that neuron #1 may receive more synaptic inputs than neuron #2 (Fig. 3e).

**Example of 2-SPARSE for neurons with different functional features**. In the experiments described above, two neurons in the same animal with similar response features were targeted by electroporation (steps 1 and 2). Next, we provide a demonstration of the use of 2-SPARSE for two neurons with distinct response features. In this example, we targeted two neurons (neurons #3 and #4 in Fig. 4a) that exhibited distinct response patterns to pure-tone stimulation, as indicated by the analyses of FRA and BF (Fig. 4b, c). We reconstructed their axonal projections using the same protocol for fMOST imaging followed by neuronal reconstruction and analysis (Fig. 4d, e). In this example, neurons #3 and #4 exhibited a similar ipsilateral-to-contralateral projection pattern, with main axons that passed through the corpus callosum and formed dense terminal fields at the contralateral AUD (Fig. 4f). Some minor differences between these two neurons included their different projection strengths in specific brain regions, such as the TEa and supplemental somatosensory area (Fig. 4e), and the location of the neuron #3 soma was deeper than that of neuron #4 (Fig. 4g). Similarly, we also quantitatively analyzed the length and a total number of branches for both the axons and dendrites of the two neurons. Interestingly, although neurons #3 and #4 exhibited distinct response features, their brain-wide projection patterns were very similar, in contrast to the previous example in which neurons #1 and #2 were functionally similar but morphologically distinct (Fig. 4h). Data from these example neurons are consistent with previous findings obtained using bulk labelling methods for L2/3 cortical neurons[28,29] in which a fraction of L2/3 sensory cortical neurons exhibited long-range interhemispheric projections. However, the lack of interhemispheric projection of neuron #2 raised the important question of whether or not this morphology was due to incomplete labelling. We, therefore, addressed this issue in the following set of experiments.

**Evaluation of the quality of 2-SPARSE for long-range projection labelling**. We evaluated the axonal labelling quality achieved by 2-SPARSE through an independent set of experiments involving long-range retrograde labelling, which is an acknowledged standard established previously[12]. The long-range projection between bilateral AUDs was considered a challenging test metric because this contralateral cortico-cortical projection distance is

~9.6 mm long, which is more than twice the ipsilateral cortico-striatal projection of 4.7 mm and is one of the longest possible projections for L2/3 neurons in the mouse neocortex (Fig. 5a). First, we performed bulk retrograde labelling with AAV2/2-Retro expressing a red fluorescent protein (mRuby3) starting from L2/3 of the contralateral AUD (the injection site of AAV2/2-Retro-hSyn-H2B-mRuby3-WPRE-pA is indicated by an arrow in the left AUD in Fig. 5b). This approach resulted in the labelling of a fraction of neurons in L2/3 of the AUD on our single-neuron targeting side (the right AUD in Fig. 5b). Then, by 2-SPARSE, we targeted a few neurons whose long-range interhemispheric contralateral projections were verified by mRuby3 signal following AAV2/2-Retro virus injection into the contralateral AUD (the two targeted neurons are shown in Fig. 5c, d). At thirty days after electroporation, the axonal terminals in the contralateral AUD of our targeted neurons were clearly visible in all tested cases (Fig. 5e, f and Supplementary Fig. 8a, b; 9 neurons from 5 mice), indicating a high success rate of 2-SPARSE in filling long-range axon collaterals (see also Supplementary Video 2). In a parallel set of control experiments, we labelled L2/3 contralateral AUD-projecting neurons in the AUD that were identified by AAV2/2-Retro virus injection followed by single-cell electroporation of the Cre-GFP plasmid, as described above but without the assistance of AAV injection. For these control neurons, the axonal projections extended out of the local AUD but could not reach the confirmed destination of the contralateral AUD (Fig. 5g, h and Supplementary Fig. 8c; 9 neurons from 5 mice), suggesting that AAV injection was important for enhancing the labelling of long-range interhemispheric projections and which showed a success rate close to 100% through our combined method.

**Comparison of 2-SPARSE with conventional AAV-based labelling**. AAV-based labelling has served as a powerful tool to elucidate the hierarchies of neural circuits due to its extraordinary properties, such as a high infection efficiency and relatively low disturbance to host cells. Previous studies[13,30] have employed a binary AAV expression system in which a diluted Cre-expressing AAV was used in combination with another AAV that expressed Cre-responsive elements (e.g., DIO) to label a limited number of neurons in the cerebral cortex (see Methods for details). To evaluate the quality of 2-SPARSE single-neuron labelling by comparison with that of the binary AAV expression system, we reproduced the binary AAV expression method in the AUDs. We reconstructed the morphologies of 6 neurons (neurons #23–28) labelled with the binary AAV expression system (Fig. 6a). The cell bodies of these 6 neurons were all located in L2/3 (2 neurons in L2 and 4 neurons in L3) (Fig. 6b). We compared these neurons with the previous set of 13 neurons (neurons #1–13) labelled and reconstructed by 2-SPARSE, as mentioned above (Figs. 3–5), by analyzing the length and total number of branches for both axons and dendrites. We found no significant differences in dendrite length ($P = 0.8314$), a total number of dendrite branches ($P = 0.3556$), axon length ($P = 0.4155$), or the total number of axon branches ($P = 0.4524$) between groups labelled by these two different methods (Fig. 6c). These results demonstrate that the quality of single-neuronal labelling using 2-SPARSE is comparable to that of the currently prevalent binary AAV expression system.

**Extended applications of 2-SPARSE in the motor cortex and its deeper layers**. To further demonstrate the application of 2-SPARSE, we performed an additional set of experiments in the motor cortex, where a rich pool of axonal projection patterns was recently revealed by a high-throughput study[13]. After single-cell plasmid electroporation and local AAV injection, we performed

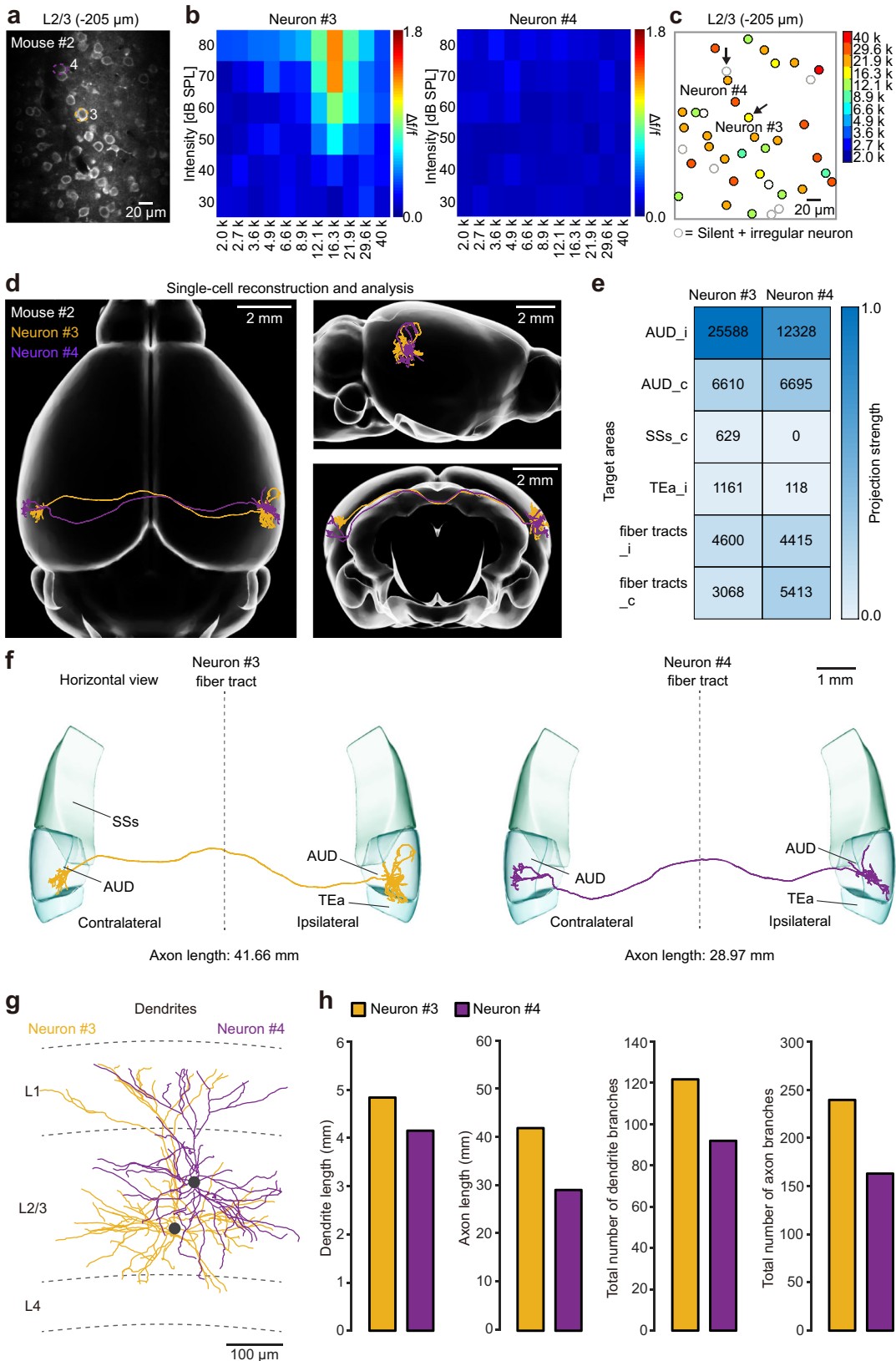

fMOST imaging and reconstructed eight intratelencephalic (IT) neurons (neurons #29–36) from two brain samples. We then registered these neurons and aligned them to the standard Allen Brain Atlas (Fig. 7a). With brain-wide axonal reconstruction and quantitative analysis, we found that these eight neurons exhibited highly diverse projection patterns, which were reflected by the

projection strengths of axon targets in diverse brain regions (Fig. 7b) as well as by the axonal morphology of each neuron (Fig. 7c).

In this dataset, there were four neurons in layer 5 (L5) and four neurons in L2/3 (Fig. 7c, d). The projection patterns of L5 neurons were more complex than those of most L2/3 neurons, as

**Fig. 4 Whole-brain reconstruction of single functionally distinct neurons. a** A representative two-photon image showing L2/3 neurons of the AUD of a head-fixed awake mouse. **b** Colour-coded FRAs of the two neurons marked in (**a**). **c** Best frequency map of all the neurons in the imaging plane shown in (**a**). The colour code is on the right. **d** Two reconstructed neurons (neurons #3 and #4) registered to the standard Allen Brain Atlas are shown in horizontal (left), sagittal (upper right), and coronal (lower right) views. **e** Projection strengths of the two neurons. The colour code reflecting the projection strength is on the right. **f** The two reconstructed neurons are displayed separately. The target areas are coloured as indicated. **g** The dendrites of the two brain-wide reconstructed neurons are displayed in different colours. The grey dashed lines indicate laminar borders. The two neuronal somata are located in L2/3. **h** Comparisons of the dendritic and axonal lengths and total numbers of axonal and dendritic branches between neuron #3 and neuron #4. AUD auditory areas; SSs supplemental somatosensory area; TEa temporal association areas.

reflected by the quantitative analysis of axon length ($P = 0.0286$, two-sided Wilcoxon rank-sum test) (Fig. 7e). The total axon length of the representative L5 neurons (neuron #34) reached as high as 212 mm, which was comparable to the axon length of representative L5 neurons (over 180 mm) reported by several previous studies[13,31]. This finding indicates that 2-SPARSE can reliably label individual neurons and enable complex reconstruction of neurons in the motor cortex.

## Discussion

Here, we developed 2-SPARSE, a method that combines two-photon $Ca^{2+}$ imaging in awake mice, targeted plasmid electroporation, and local viral injection guided by two-photon imaging, post-mortem whole-brain sectioning and imaging, and morphological tracing. With 2-SPARSE, we were able to precisely label single neurons with distinct sound frequency tuning features in the AUD of awake mice (Fig. 1) and reconstruct long-range axonal projections that spanned the entire brain (Figs. 4, 5). 2-SPARSE enables 100% accuracy in matching brain-wide morphology with selected functional features for individual neurons. A key feature of this method is its combination of single-cell electroporation of a Cre-expressing plasmid with close, local injection of AAV injection to enable Cre-mediated activation and amplification of robust fluorescence expression in targeted neurons. This approach provides higher accuracy than that of conventional methods that are based only on conventional filling of cells with biocytin or EGFP plasmids[32–34]. When we compared our findings with the mesoscale projection patterns from the Allen Mouse Brain Atlas, we found that the single-cell target brain regions we identified in the AUD were highly concordant with those of the mesoscale data[3,11]. This finding suggests that the high-fidelity of 2-SPARSE supports its use in cross-validation of results obtained by high-throughput methods that have high variability among individual cells in the integrity of labelling.

Compared with widely used high-throughput approaches that primarily involve viral vector-mediated gene transfer[13,17,35] with uncontrollable randomness in cell targeting, 2-SPARSE has a unique advantage in that it enables precisely targeted labelling of single, functionally defined neurons after functional population imaging (Fig. 1). This precision is particularly useful when the functionally defined neurons in a specific brain region are sparse in number but exhibit wide spatial distribution and are intermingled with other neurons in the same region. In recent decades, many such neuronal types have been identified across multiple brain regions. For example, feature-specific neurons have been identified in the AUD[6] and visual cortex[36] for processing of sensory information; place cells[37] and grid cells[38] have been identified in the hippocampus and the entorhinal cortex that participate in spatial navigation; engram cells[39–41] have been identified in the hippocampus and cortex for memory processing, and holistic bursting cells[10] have been identified in the AUD for complex sound encoding. These types of functionally defined cells constitute just a small proportion of the neuronal population in specific brain regions (e.g., grid cells, ~20%; engram cells, ~4%; and holistic bursting cells, ~5%) and are inevitably spatially

intermingled. Moreover, these functional neuron types largely lack known molecular markers. Thus, molecular approaches alone[42–45] are frequently limited in their potential application for mapping between functional and morphological features. In this case, 2-SPARSE not only provides an ideal solution for acquiring precise data but can also serve as an annotation reference for benchmarking and calibration of high-throughput population mapping methods used to correlate functional and morphological features of neurons.

It warrants mention that 2-SPARSE, although extremely precise, has several drawbacks. First, the workflow requires intensive real-time interactive operations. However, once well established and practised, an overall success rate of >80% can be achieved in consecutive experiments (Supplementary Tables 2–5). Notably, the success of each step can be readily confirmed by visualization before proceeding to the next step, thus allowing the valuable experience to be gained after short cycles of troubleshooting. Second, visual guidance by two-photon imaging is required for precise single-cell targeting; thus, 2-SPARSE is limited to brain regions that are optically accessible by two-photon imaging with a cranial window that allows micropipette manipulation under the microscope objective. Here, we demonstrated the extended application of 2-SPARSE in L5 neurons of the mouse motor cortex (Fig. 7). In principle, 2-SPARSE could also be applied to hippocampal regions in which a combination of two-photon imaging and single-cell electrophysiology in vivo was feasible in previous reports by experienced researchers[46–48], although analysis of this region is likely to be more difficult. Third, to avoid optical interference, we did not use fluorescent $Ca^{2+}$-sensitive proteins for two-photon imaging but rather used chemical $Ca^{2+}$ dyes that faded after 1 day. The use of these dyes may be a limitation for studies that are aimed at chronic tracking of changes in the functional features of individual cells[10]. A possible solution to address this limitation is to employ $Ca^{2+}$-sensitive proteins and morphology indicator proteins with different colours.

Implementation of 2-SPARSE does not require new instruments or reagents beyond those commonly used in neuroscience research, but requires the effective combination of these instruments, as demonstrated in this study. The aspects of this technique that require the most care and expertize are the integration of a conventional micropipette manipulator and an electroporation amplifier into a standard two-photon microscope with video-rate full-frame imaging capability. In our experience, considerable training and practice are essential to simultaneously operate these devices in one experiment. We anticipate that the successful implementation of 2-SPARSE will enable major breakthroughs in brain connectomics through this advance in single-neuron functional projectomics, thus bridging neuroanatomy and neurophysiology with single-cell precision in vivo.

## Methods

**Animals.** All experimental procedures related to the use of animals were approved by the Third Military Medical University Animal Care and Use Committee and were carried out in accordance with institutional animal welfare guidelines. Adult (8- to 12-week-old) male C57BL/6J mice were obtained from the Laboratory Animal Centre at the Third Military Medical University. The mice were given

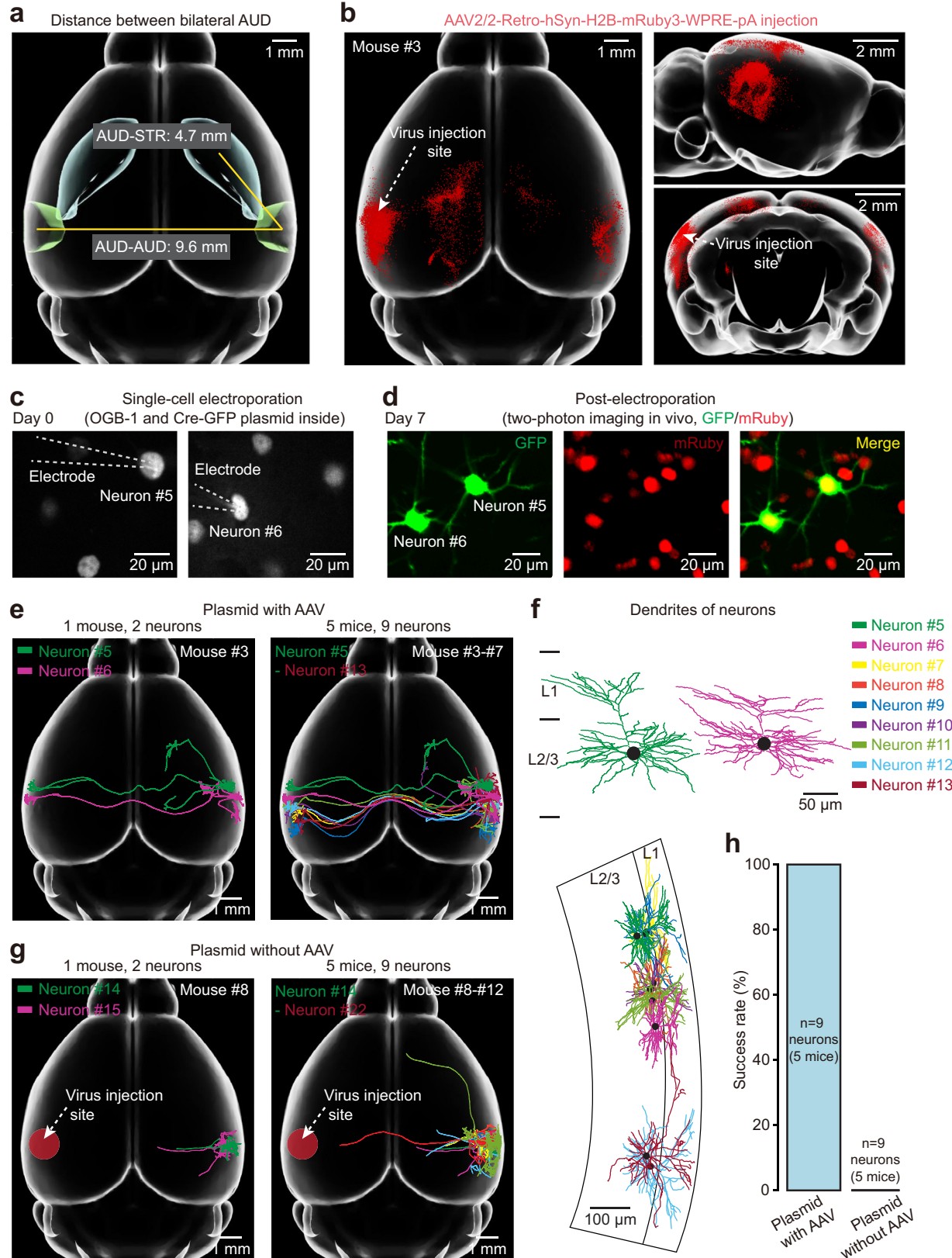

**a** Distance between bilateral AUD

AUD-STR: 4.7 mm
AUD-AUD: 9.6 mm

**b** AAV2/2-Retro-hSyn-H2B-mRuby3-WPRE-pA injection

Mouse #3
Virus injection site

**c** Single-cell electroporation
(OGB-1 and Cre-GFP plasmid inside)
Day 0
Electrode
Neuron #5
Electrode
Neuron #6

**d** Post-electroporation
(two-photon imaging in vivo, GFP/mRuby)
Day 7
GFP    mRuby    Merge
Neuron #5
Neuron #6

**e** Plasmid with AAV
1 mouse, 2 neurons
Neuron #5
Neuron #6    Mouse #3

5 mice, 9 neurons
Neuron #5
- Neuron #13    Mouse #3-#7

**f** Dendrites of neurons
L1
L2/3
— Neuron #5
— Neuron #6
— Neuron #7
— Neuron #8
— Neuron #9
— Neuron #10
— Neuron #11
— Neuron #12
— Neuron #13

L2/3    L1

**g** Plasmid without AAV
1 mouse, 2 neurons
Neuron #14
Neuron #15    Mouse #8
Virus injection site

5 mice, 9 neurons
Neuron #14
- Neuron #22    Mouse #8-#12
Virus injection site

**h**
Success rate (%)
Plasmid with AAV
n=9 neurons (5 mice)
Plasmid without AAV
n=9 neurons (5 mice)

access to food and water *ad libitum*, and they were housed in a humidity-(40–50%) and temperature-(20–22 °C) controlled room with a 12 h light/dark cycle (lights off at 19:00).

**Auditory stimulation**. Auditory stimuli were delivered through a free-field ES1 speaker using an ED1 electrostatic speaker driver (Tucker Davis Technologies, USA). The speaker was placed ~6 cm from the mouse ear that was contralateral to the recorded AUD[49–51]. The auditory stimuli were generated by a custom LabVIEW 2016 programme (National Instruments, USA) and converted to analogue voltages with a PCI6731 card (National Instruments, USA). A pre-polarized condenser microphone (377A01 microphone, PCB Piezotronics Inc., USA) was used to calibrate the generated sounds. To establish the FRA of each neuron, sequences of pure tones (50 ms duration)

**Fig. 5 Axons projecting to the contralateral AUD can be reliably labelled and reconstructed. a** The distance between the ipsilateral and contralateral AUDs is twice as long as that between the ipsilateral AUD and striatum. **b** Brain-wide distributions of AUD-projecting neurons achieved by a retrograde labelling strategy (AAV2/2-Retro-mRuby3). The detected somata were registered to the standard Allen Brain Atlas, as denoted by red dots. The white dashed arrow indicates the injection site. Three different views are shown (left: horizontal; upper right: sagittal; lower right: coronal). **c** Demonstration of single-cell electroporation. **d** Representative example showing two-photon imaging of two dual-colour labelled neurons (neurons #5 and #6) on day 7 after electroporation. **e** Left, reconstruction of two representative neurons (neurons #5–6; obtained from one mouse) labelled with a plasmid together with local AAV injection. Right, reconstruction of nine neurons (neurons #5–13; obtained from five mice) labelled with a plasmid with local AAV injection.
**f** Reconstructed dendrites of the two representative neurons (top) in (**e**) (left) and all nine neurons (bottom) in (**e**) (right). **g** Left, reconstruction of two representative neurons (neurons #14–15; obtained from one mouse) labelled with a plasmid without nearby AAV injection. Right panel, reconstruction of nine neurons (neurons #14–22; obtained from five brains) labelled with a plasmid without nearby AAV injection. **h** Comparison of the success rates of axonal terminal filling in the contralateral AUD with a plasmid with AAV injection and without AAV injection (*n* = 9 neurons from 5 mice for each group).

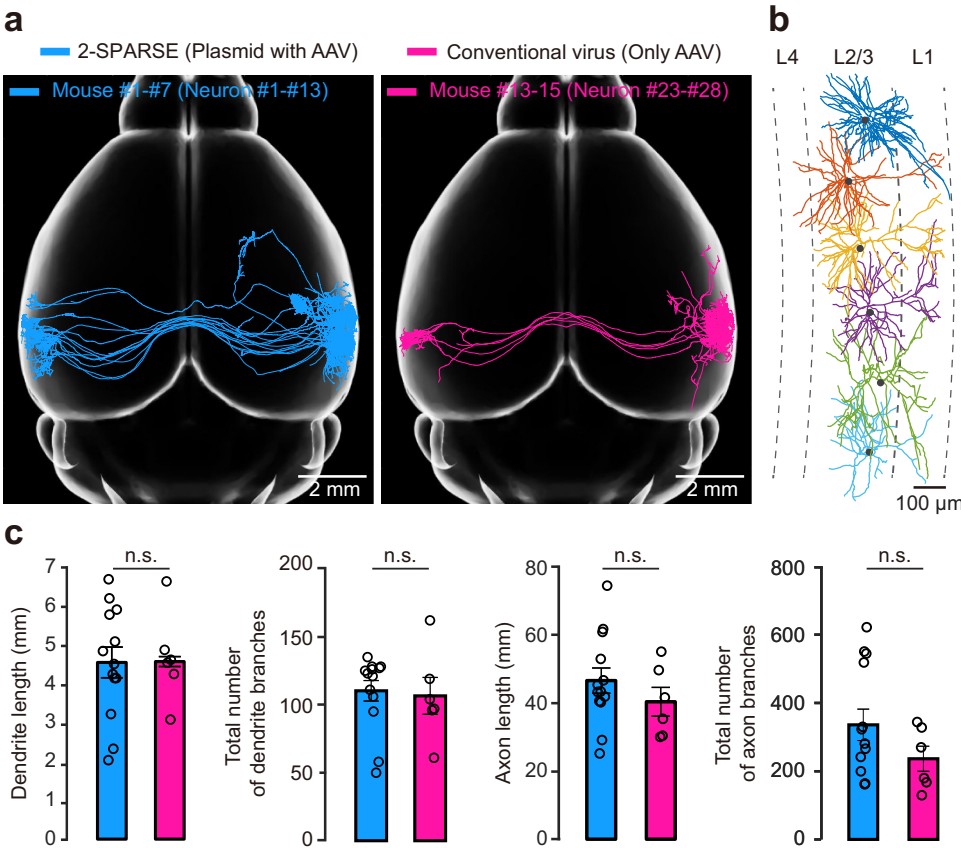

**Fig. 6 Comparisons of efficacy between 2-SPARSE and a conventional virus expression system in the AUD. a** All reconstructed neurons labelled by 2-SPARSE (neurons #1–13; cyan) or a conventional virus expression system (neurons #23–28; pink). The neurons were registered to the standard Allen Brain Atlas and are shown with horizontal view. **b** Dendritic morphologies of five reconstructed neurons (corresponding to the pink neurons in (**a**) labelled with a conventional virus. All neurons are located in the L2/3 and displayed in different colours. The grey dashed lines indicate the laminar borders.
**c** Comparisons of dendritic and axonal lengths and total numbers of axonal and dendritic branches between the neurons labelled by 2-SPARSE (*n* = 13 neurons) and the conventional virus expression system (*n* = 6 neurons). The data with error bars represent the mean ± SEM. *P* = 0.8314, *P* = 0.3556, *P* = 0.4155, and *P* = 0.4524 for the dendritic length, dendrite branch number, axon length, and axon branch number comparisons, respectively; two-sided Wilcoxon rank-sum test, n.s., *P* > 0.05.

consisting of 5 sound levels (30–80 dB sound pressure level (SPL), 10 dB attenuation), and 11 frequencies (logarithmic scale from 2 to 40 kHz) were presented with 8 repetitions and randomized intervals of 2–4 s. During auditory stimulation, no other sensory stimulus was present to the mouse.

**Two-photon Ca$^{2+}$ imaging in the mouse AUD in vivo.** To conduct Ca$^{2+}$ imaging in awake mice[23], we habituated each mouse to head fixation with a titanium head post and exposed the right primary AUD under isoflurane anaesthesia. A custom-made recording chamber was attached to the skull with cyanoacrylate glue (UHU). A craniotomy (~2 mm × 2 mm) was performed above the AUD (location centre: −3.0 mm from the bregma, 4.5 mm lateral to the midline) and filled with 1.5% low-melting-point agarose. The recording chamber was perfused with normal artificial cerebrospinal fluid (ACSF) containing (in mM) 125 NaCl, 4.5 KCl, 26 NaHCO$_3$,

1.25 NaH$_2$PO$_4$, 2 CaCl$_2$, 1 MgCl$_2$ and 20 glucose, and the pH was 7.40 when equilibrated with 95% oxygen and 5% CO$_2$. To perform bolus loading, Cal-520 AM[52] at a concentration of 567 μM was obtained by dissolution in DMSO with 20% Pluronic F-127. A puller (PC-10; Narishige, Tokyo, Japan) was used to make a borosilicate glass micropipette for dye injection with the pulling mode set as two-stage and a heavy type. The injection pipette was filled with normal ACSF, and its tip maintained a resistance of 2 MΩ. The injection pipette was controlled to approach the area of interest with a pressure of 30 mbar and to inject dye solution with a pressure of 600 mbar (3 min). Two hours after dye injection and complete removal of isoflurane anaesthesia, Ca$^{2+}$ imaging was performed. A custom-built two-photon microscope (LotosScan 1.0, Suzhou Institute of Biomedical Engineering and Technology, Chinese Academy of Sciences, China)[53] was used to perform two-photon Ca$^{2+}$ imaging experiments. Excitation light of 920 nm wavelength was delivered to the brain with a Ti:Sa laser (power of 30–120 mW,

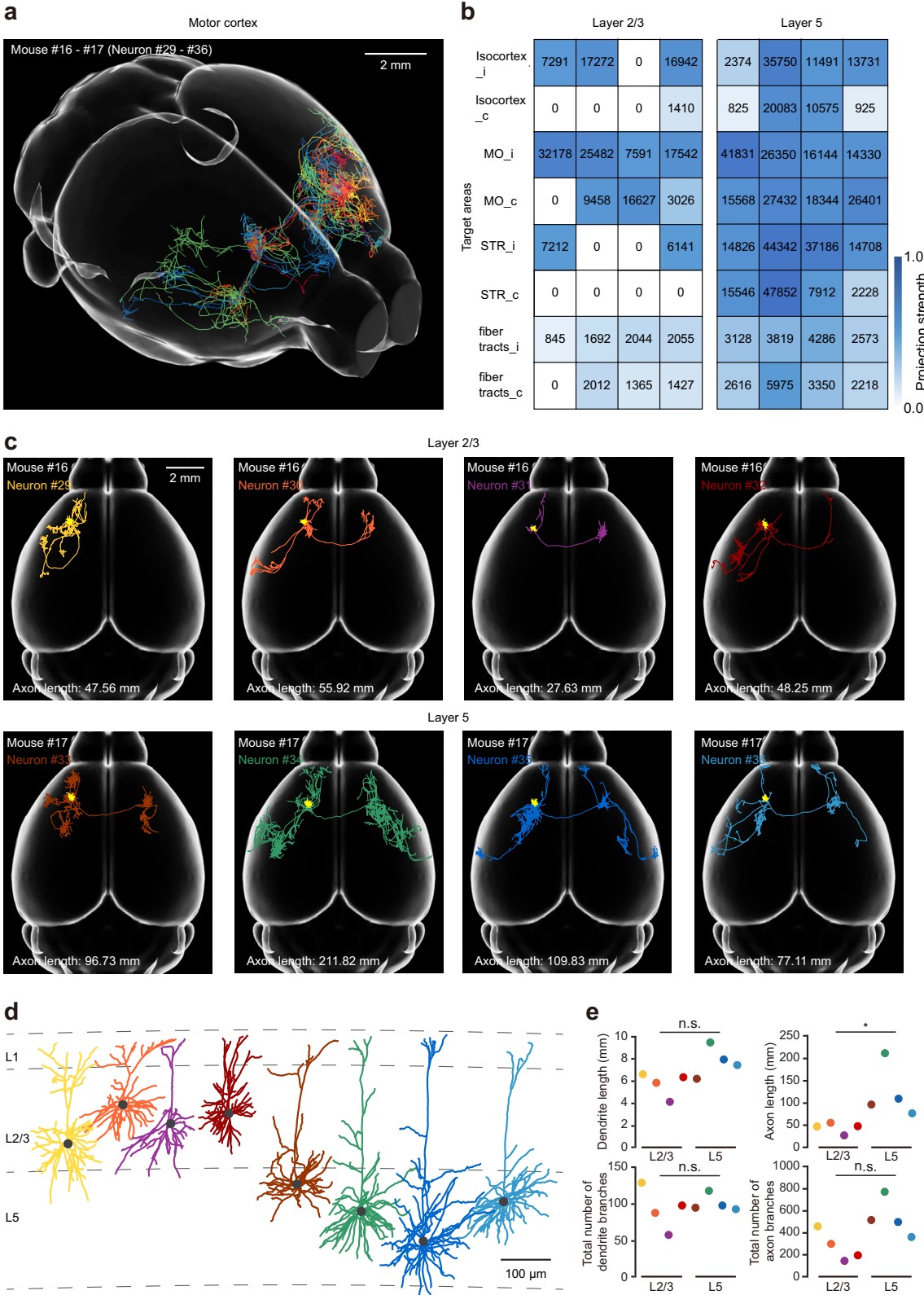

**Fig. 7 Labelling and reconstruction of IT neurons in the motor cortex by 2-SPARSE. a** Three-dimensional visualization of IT neurons in the motor cortex. **b** Projection patterns of all the reconstructed IT neurons. The columns represent individual IT neurons. The colour code is shown on the right side. **c** Horizontal view of the complete morphology of individual IT neurons with different colour coding. Dendrites are shown in yellow. **d** Dendrite morphologies of the reconstructed neurons in (**c**). The grey dashed lines indicate laminar borders. **e** Comparisons of dendritic ($P = 0.1143$) and axonal lengths ($P = 0.0286$) and the total numbers of dendritic ($P = 0.7429$) and axonal branches ($P = 0.0571$) between the reconstructed L2/3 neurons and L5 neurons in the motor cortex. $n = 4$ neurons for both cases, two-sided Wilcoxon rank-sum test, n.s. $P > 0.05$, *$P < 0.05$.

locked mode, Mai-Tai DeepSee, Spectra Physics) and a water-immersion objective (40×, 0.8 NA, Nikon Corporation). Two-photon images (600 pixels × 600 pixels) were recorded at a 40 Hz frame rate.

**Data analysis for Ca$^{2+}$ transients**. We analyzed the Ca$^{2+}$ imaging data with custom-written software in MATLAB 2018b (MathWorks)[54]. Individual neurons were visually/semiautomatically identified and segmented as regions of interest (ROIs) according to cell morphology and fluorescence intensity. Fluorescence values ($f$) across time for each neuron were obtained by averaging the intensity values corresponding to the pixels within each ROI for each imaging frame. For each neuron, the baseline fluorescence $f_0$ was estimated as the 25th percentile of the entire fluorescence recording, and then Ca$^{2+}$ signals were calculated as relative fluorescence changes $\Delta f/f = (f − f_0)/f_0$. Detection of Ca$^{2+}$ transients was carried out based on an integrated thresholding approach used in our previous studies[51]. The FRA of each neuron was constructed by calculating the responses for the defined frequency-intensity conditions.

**Two-photon-targeted plasmid electroporation**. To label a single neuron[12] in the AUD (location centre: −3.0 mm from the bregma, 4.5 mm lateral to the midline) or the motor cortex (location centre: +1.5 mm from the bregma, 1.6 mm lateral to the midline), a patch pipette (10–12 MΩ resistance) containing intracellular solution, OGB-1-6K$^+$ (100 μM; Invitrogen) and plasmid DNA (100 ng/μl of hSyn-eGFP-P2A-Cre-pA) was placed over the craniotomy within the microscope's field of view. The electroporation pipette was advanced through the dura, while high positive pressure pulses (>100 mbar) were applied to the back of the pipette with a syringe. The pipette was then inserted into the brain and advanced to L2/3 or L5 with a reduced pipette pressure of 50 mbar. Positive pressure was applied to the back of the pipette to fill the extracellular space with dye. A cloud of fluorescent dye surrounding the pipette was visible in the brain. Single neurons and the tip of the pipette were identified using shadow imaging. In some cases, high positive pressure was applied for 3–5 s to clean the tip, and then we continued to search for neurons under low positive pressure (~30–50 mbar). Once the pipette was in close proximity to the targeted neuron, the pressure was lowered to approximately 10 mbar. The pipette was slowly advanced towards the centre of the soma of the neuron. The positioning of the pipette close to the neuron membrane was crucial for successful filling. After the tip of the pipette was in contact with the neuron membrane, it was moved very gently to the centre of the cell body, with a mean distance of 2 μm. A small dimple filled with dye was observed around the pipette tip in the neuron membrane. At this moment, the pressure on the pipette was released. Then, electroporation pulses (NPI Electronic, Germany) were delivered by an MVCS-01 iontophoresis system. The parameters of the electrical pulses were as follows: a pulse amplitude of −450 nA (note the negative polarity, which should be set on the device), a single pulse duration of 500 μs, and a train of 100 pulses at 50 Hz pulsing frequency were applied.

Successful electroporation was verified by immediately filling the neuron body with dye. After verification, the pipette was slowly withdrawn and exchanged for electroporation of another neuron. To minimize damage to the cortex, it is critical to avoid making very large lateral or vertical pipette movements within the brain. In our experiments, we did not move the pipette more than 50 μm laterally or vertically within the brain. Spontaneous activity was recorded for several minutes to ensure that the electroporated neuron was still alive. In consecutive experiments, more than 90% of neurons remained functional after electroporation (Fig. 2 and Supplementary Table 2).

**AAV injection and craniotomy sealing**. After electroporation of one or a few neurons, the electroporation pipette was retracted. Then, an injection pipette (2 MΩ resistance) containing intracellular solution, OGB-1-6K$^+$ (10 μM, Invitrogen, or Alexa Fluor 594, 50 μM), and AAV-DIO-mGFP was placed over the craniotomy within the microscope's field of view. The pipette was advanced through the dura with high positive pressure pulses (>100 mbar). The pipette was then inserted into the brain and advanced to L2/3 with a reduced pipette holding pressure of 50 mbar. After the tip of the pipette was approximately 30 μm away from the electroporated neuron, the holding pressure was released on the pipette. Then, a high positive pressure (~300 mbar, 2 min) injection was applied near the location of the electroporated neurons. No more than nine penetrations were conducted per mouse.

Following virus injection, the craniotomy was covered with two pieces of coverslip glass and sealed. A small (1.5 mm diameter) coverslip was positioned below and a large (2.5 mm diameter) coverslip was positioned above the craniotomy to cover the brain and part of the skull, respectively. Afterwards, dental acrylic and ultraviolet-cured optical adhesives (Norland Products Inc., USA) were used to seal the skull. The mice were carefully monitored during recovery. All mice showed normal behaviour, with no signs of distress. Successful recovery of the electroporated cells was determined by two-photon imaging through the cranial glass window for five days after the electroporation day. For this test imaging, mice were briefly anaesthetized with isoflurane.

**Retrograde tracing (control experiment)**. To label the contralateral AUD-projecting neurons located in the AUD, we injected a viral solution (AAV2/2-Retro-hSyn-H2B-mRuby3-WPRE-pA) into the contralateral AUD with a patch

pipette (resistance 2 MΩ)[12]. The viral solution was loaded by applying pressure (~300 mbar, 2 min) to the pipette. Fourteen days after viral injection, the somata of labelled neurons were observed in the AUD by two-photon imaging in vivo. A subgroup of the labelled auditory neurons was chosen for electroporation, as shown in Fig. 5.

**Neuronal sparse labelling with AAVs (control experiment)**. To achieve neuronal sparse labelling, i.e., labelling of a limited number of neurons in the AUD, we performed virus injection with an AAV-mediated binary expression system based mainly on previously reported protocols[13,30]. Briefly, a Cre-expressing AAV (AAV2/8-hSyn-Cre-WPRE-pA) was diluted with 0.01 M phosphate-buffered saline (PBS) at a 1:10000 ratio and then mixed with an equal volume of the other Cre-responsive AAV (AAV2/9-hSyn-DIO-mGFP-WPRE-pA). The 0.01 M PBS contained 0.138 M NaCl and 0.0027 M KCl, pH 7.40 as measured by a pH metre, accurate to two decimal places (Sigma-Aldrich, cat. no. P3813-10PAK). A 100 nl total volume of virus mixture was injected into the AUDs of wild-type mice with the following coordinates: −3.0 mm from the bregma and 4.5 mm lateral to the midline according to The Mouse Brain in Stereotaxic Coordinates[55].

**Plasmid and AAVs**. The plasmid used in this study for single-cell electroporation was Cre-GFP (hSyn-eGFP-P2A-Cre-pA, 100 ng/μl). The plasmid was purchased from Genscript Co., Ltd. (Nanjing, China). The AAVs used in this study were AAV2/8-hSyn-Cre-WPRE-pA (AAV2/8, titre: $1.30 \times 10^{13}$ viral particles/mL), AAV-hSyn-DIO-mGFP-WPRE-pA (AAV2/9, titre: $2.59 \times 10^{12}$ viral particles/mL) and AAV-Retro-hSyn-H2B-mRuby3-WPRE-pA (AAV2/2, titre: $2.59 \times 10^{12}$ viral particles/mL). All AAVs used in our experiments were obtained from Taitool Bioscience Co., Ltd. (Shanghai, China).

**fMOST imaging**. In brief, brain samples were first dissected, postfixed in 4% paraformaldehyde at 4 °C for 24 h, rinsed with 0.01 M PBS and embedded in Lowicryl HM20 (Electron Microscopy Sciences, cat. no. 14340) before the imaging experiment. Afterwards, the resin-embedded brains were transferred to an fMOST system (BioMapping5000, Wuhan OE-Bio Co., Ltd.)[56,57] with an imaging resolution of $0.3 \times 0.3 \times 1$ μm$^3$ and imaged in a water bath filled with propidium iodide (PI). The entire brain sample was imaged (fMOSTViewer, version 1.0) in the coronal plane for both the GFP and PI channels and then sliced to 1 μm or 2 μm with a diamond knife. The cycle of brain imaging and sample sectioning was conducted continuously until data acquisition was complete[58].

**Single-cell reconstruction**. To improve the signal-to-noise ratio of the fMOST imaging data obtained from both the GFP and PI channels, image preprocessing procedures were performed that included image stitching, brightness adjustment, and noise filtering. The preprocessed imaging data were saved for both the PI and GFP channels. We transformed the preprocessed data into cuboid data via TDat 2017 software[59]. After importing the data into Amira software (version 6.1.1, FEI), we viewed the image stacks and traced the neurite skeletons in an interactive manner using the filament editor[17]. Two experienced annotators traced each neuron independently and then compared their reconstructions to produce a final consensus. To make direct comparisons between the morphologies of L2/3 long-range axonal projections labelled by 2-SPARSE and those labelled by the conventional binary AAV expression system, we randomly selected axon branches in contralateral cortices as starting points and performed manual neuronal tracing in retrograde directions, that is, from the contralateral hemispheres to the somata in the injection site, following the same tracing procedures for both groups. For neuronal reconstruction in all other cases, tracing was performed in anterograde directions (from the somata in the injection site to diverse target regions). All traced points for reconstructing neurons were saved in SWC format.

**Image registration and quantitative analysis**. The whole-brain imaging data were registered to a 3D reference brain atlas called the Allen Common Coordinate Framework (CCF)[60]. The imaging data recorded from the PI channel were downsampled to a spatial resolution of $10 \times 10 \times 10$ μm$^3$. First, the imaged brains were corrected with rigid registration. A greyscale-based 3D affine registration was used for alignment to the Allen Brain Atlas, after which dense landmark-based 2D registration was performed in local regions. All registration processes were performed using Elastix[61]. Subsequently, the traced points of neuron reconstructions were transformed into Allen Brain Atlas space using the parameters for the registration transformation. Two experienced analysts checked the image registration results by back-to-back manual confirmation. Morphological features were quantitatively analyzed using Amira and custom-written software in MATLAB. The branch lengths for both axons and dendrites were measured by summing the distances from the traced points to their parent nodes. The branch numbers for both axons and dendrites were measured by counting the segments that connected to the parent nodes.

**Statistics and reproducibility**. In the figurest, summarized data are presented as the mean ± SEM. To compare data between groups, we used non-parametric Wilcoxon rank-sum test (unpaired) and Wilcoxon signed-rank test (paired) to

determine statistical significance between them. For the representative experiments in Figs. 1b, e, 4a, 5c, d; Supplementary Figs. 2, 3a, b, 4a–d, 5b, c, these experiments were repeated independently, and similar results were obtained from n = 2, 7, 2, 5, 4, 7, 7, and 7 mice, respectively.

**Reporting summary**. Further information on the research design is available in the Nature Research Reporting Summary linked to this article.

## Data availability

Raw unprocessed data have a large size (>100 TB), but the original data that support the findings of this study are available from the corresponding author upon request. Source data underlying Figs. 1–7 and Supplementary Fig. 1 are available as a Source data file. Source data are provided with this paper.

## Code availability

The code supporting the current study has not been deposited in a public repository, but it is available from the corresponding author upon request.

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

## Acknowledgements

The authors are grateful to Drs. Yi Zhou, Xiaojun Wang, and Anan Li for technical support; to Ms. Jia Lou for help in composing and layout editing of the figures; to Profs. Hui Gong and Qingming Luo for comments. This study was supported by grants from the National Natural Science Foundation of China to X.C. (No. 31925018, 31921003, 32127801, 31861143038), X.L. (No. 32171096), and S.Z. (No. 61721092), the 973 Programme (2015CB755603 and 2015CB755602) to S.Z., Chongqing Basic Research grants (nos. cstc2019jcyjjqX0001 and cstc2019jcyj-cxttX0005) to X.C., the Key Scientific Research Equipment Development Project of the CAS (Super-resolution Microscopy Systems and Key Components, ZDYZ2013-1) to H.J., and the "100-Talents Programme for Elite Engineers" of the CAS (H.J.).

## Author contributions

X.C., H.J., S.Z., and X.L. conceived the project. X.C., M.W., K.L., and X.L. designed the experiments; M.W., K. L., J.P., J.L., P.S., Y.Z., L.L., W.G., Q.X., Z.Z., Y.L., Z.Z., J.L., T.Z., S.Z., X.L., and X.C. performed the experiments; M.W., J.L., Y.Z., C.Z., S.Z., X.L., H.J., and X.C. devised the data analysis methods; M.W., K.L., J.P., J.L., P.S., Y.Z., Q.X., Z.Zhou, Y.H., S.Z., C.Z., X.L., H.J., and X.C. performed the data analysis; S.Z., X.L., H.J., and X.C. inspected the data and evaluated the findings; X.L., H.J., and X.C. wrote the manuscript with the help from all authors.

## Competing interests

The authors declare no competing interests.
