## [Peer review file · Nature Communications]

REVIEWER COMMENTS

Reviewer #1 (Remarks to the Author):

This reviewer carefully read the manuscript "Brain-wide projection reconstruction of single functionally defined neurons" by the authors. The manuscript introduces a novel method to connect in vivo neural function with morphology information (neural projections). The authors have demonstrated the scheme to label sound-evoked neurons observed by two-photon Ca²⁺ imaging by targeting techniques (single-cell plasmid electroporation and AAV) and obtain several neuronal projection data by high-resolution imaging at whole-brain scale (fMOST). This method contributes to connecting physiological function along with neural projection, and this data is the novel part of this paper. Overall, the reviewer was convinced that the manuscript would be valuable to the journal and tries to suggest some points to be improved for the publication.

Major concerns:

#1. (Page 24, Line 398) The description of "0.01M PBS" is unclear. If the author used modified PBS, please describe the concentration of each of the components and pH.

Minor concerns:

#1. (Page 1, Line 19) The section of "Main text" would be better if divided into some conventional subjects (Introduction, Results, Discussion).

Reviewer #2 (Remarks to the Author):

This is a very interesting ms, which describes a method for morphological and functional phenotyping of single neurons in vivo. The method combines 2-photon calcium imaging in awake head-fixed mice for functional definition of single cells in a region of interest (e.g. sound-evoked responses) with single cell plasmid electroporation to delineate brain-wide projections in a cell specific manner using subsequent optical tomography. The paper illustrates the method applied to the study of the auditory cortex and describes a number of observations. The approach is outstanding and it may facilitate a better morphological profiling of brain functional granularity.

Major comments:

1- A major issue for successful dissemination of single-cell methods is feasibility. Most of the time the method works in the expert hands, but acquiring expertise may be challenging. While this is inherent to the method itself, providing a realistic view is essential. The ms will benefit from including specific assessment of the method feasibility in terms of the proportion of recovered cells from the total number of attempts. I would even consider adding a figure or a table to reinforce the critical milestones and the associated rate of success. Also, it may be important to clarify how many neurons are advisable to attempt to avoid issues with posterior morphological reconstruction in terms of separability.

2- The scientific/research component of the approach should be reinforced. While authors provide examples and some quantification, the ms will benefit from reinforcing statistical report. The number of mice and experimental design should be more clearly stated in the text. Referring to neuron1 or 2 across figures is confusing as they seem to reflect different mice and experiments (for instance, dendritic density of neurons in fig.2d and sup.fig.5 look quite different). Questions emerge regarding any main statistical trend for neurons projecting to similar targets versus those splitting apart, or any other dimension which reduces variability. It looks like Fig.2 reflects the result of only one experiment, but supplementary figures suggest there may be some additional data. Can authors converge at some main scientific message?

3- The manuscript reads as a continuous piece. However, Introduction and Results will be better organized in separated sections. Also, a more formal Discussion section may help to put the method in context, highlighting novelty and comparison with existing solutions. Importantly, the method is limited in terms of the yield. It may be argued that variability of single-cell data will prevent to make conclusions. However, there is value in adopting this approach which could eventually permit annotation of alternative high-throughput approaches based on the much fewer but morphologically and functionally identified neurons. Authors should consider these dimensions along a more detailed discussion.

Other comments

4- Some aspects of the method should be better explained. For instance, electroporation is a critical state. Are authors monitoring the cell response? How do they know a cell is successfully electroporated? Is the intracellular dye used for this purpose? How much is the pipette advanced towards the soma? Is there any potential issue with viral spilling over from the pipette? How many cells remain functional after electroporation and can be tracked across days?

5- Some statements should be clarified: line 73 “complete and robust?”; line 116 “bilateral hemispheres?”; line 143 “confirmed interhemispheric cortico-cortical projections by single-cell electroporation..?” If I understand correctly interhemispheric cortico-cortical projections are determined by AAV2Retro virus followed by single-cell electroporation of the Cre-GFP plasmid

6- Fig.2: please, clarify what are abbreviations used in panel b. Regarding this, it looks like neurons have their soma in TEa not in AUD, so strictly speaking they are neurons from the temporal cortex. Is that correct?

7- Fig.2d and e, please use the same color code. For fig.2d, how many planes are stacked per each cell? Al L1 to L4 limits comparable for the two neurons shown here? Please, add details in caption.

Reviewer #3 (Remarks to the Author):

In this manuscript, Brain-wide projection reconstruction of single, functionally defined neurons the authors attempt to tackle an important challenge in behavioral neuroscience, obtaining the combined functional characterization and neuronal morphology of individual neurons. As the authors point out, this is a technically challenging problem, especially doing this systematically or at scale. However, this manuscript does not provide any novel technological enhancements that would make such combined analysis possible routinely in a brain area of interest. Further, there are no new biological insights or conclusions from the few neurons characterized in this study. For these reasons this paper should not be published in its current form in Nature Communications.

Specific Comments:

1) Electrophysiological recordings combined with juxtacellular labeling and morphological reconstructions are not new. Multiple published works exist, some nice examples can be found in the following (Deschênes et al., 1998; Furuta et al., 2011; Li et al., 2017). The authors do not reference any earlier published work that do this. Are the authors suggesting that their 2P calcium imaging combined with targeted labeling is a higher-throughput approach? If so, this is not borne out by the low yield of reconstructed neurons in each brain or the total neurons characterized in this manner in this study.

Deschênes, M., Veinante, P., and Zhang, Z.-W. (1998). The organization of corticothalamic projections: reciprocity versus parity. *Brain Research Reviews* 28, 286–308.

Furuta, T., Deschênes, M., and Kaneko, T. (2011). Anisotropic distribution of thalamocortical boutons in barrels. *J Neurosci* 31, 6432–6439.

Li, L., Ouellette, B., Stoy, W.A., Garren, E.J., Daigle, T.L., Forest, C.R., Koch, C., and Zeng, H. (2017). A robot for high yield electrophysiology and morphology of single neurons in vivo. *Nature Communications* 8, 15604.

2) There are no new biological results that emerge that relate neuronal function and their structure. Are there neurons with similar functional properties that are structurally very different or alternatively, are there neurons with unique structural signatures that have distinct functional correlates.

3) There are two neurons shown in figure 2 that have the same tonal properties but with different morphologies, but the data is too anecdotal to draw any conclusions. More exemplars of the two morphologies would be needed.

4) The data is not convincing enough to ensure completeness of cell filling and reconstructions. For instance, is the result of the two different morphologies in figure 2 real, is the non-contralateral projecting neuron just an incomplete fill and reconstruction? A retrograde tracing experiment that is contingent on targeting both the contralateral AUDp and CP should be done to corroborate this result.

5) Providing some context for the completeness of reconstructions would be important. For instance, from comparisons to the mesoscale data from the Allen institute what is the expectation for brain areas targeted by layer 2/3 neurons from AUDp. What are expectations for number of morphological types?

6) Additional proof for completeness of cell fills by this approach could also come from indirect experiments wherein morphological reconstructions are done in AUDp cells labeled using established viral reagents showing concordance between the two methods.

7) It is not clear that the approach used in this study will generalize to other neuron types including IT neurons from motor cortex for instance that appear to be far more complex (>35cms of axonal length) than any neurons reconstructed in this study.

8) Can deeper layer cells be targeted efficiently?

We would like to express our deep and sincere appreciation to the reviewers for their constructive comments and suggestions towards improving our manuscript (NCOMMS-21-07308-T). To address their questions, we have performed several additional experiments and analyses, extensively modifying our manuscript. We have also accordingly appended further detailed discussions to the corresponding sections of the paper and provided responses to each of the individual comments. We have marked all the main changes in blue in the manuscript.

Overview of the major changes made in the revised manuscript:

- (1) A new pair of example neurons with substantially different preferred tuning frequencies were targeted, labeled and reconstructed, shown in new Figure 4.
- (2) We provide a comparison between our method and a previously reported binary AAV expression approach (sparse labelling, but no single-cell targeting) under the same experimental conditions. These new data are presented in revised Figure 6.
- (3) In an extended demonstration of our method, we show the labeling and reconstruction of single neurons in the motor cortex from layers 2/3 and layer 5, presented in Figure 7.
- (4) We have added information regarding the essential steps, problems, possible explanations, and solutions for reinforcing the critical milestones in Table 1.
- (5) The success rate of labeling has been appended to Table 2, and the spontaneous activities observed before and after electroporation have been added to Figure 2.
- (6) We added morphological parameters of the reconstructed neurons from different labelling methods to Table 3 and the target areas of the reconstructed neurons to Table 4.
- (7) We extended the discussion section to highlight the importance and novelty of our method.
- (8) We have added the requested details of single-cell electroporation to the method section.
- (9) We have addressed all other issues from each reviewer.
- (10) In addition, due to the additional work conducted at the advice of reviewers, we would like to add two more authors and change the order of authors due to their contributions during the revision process. We again thank the editor for their assistance and careful handling of our study, which we believe has been greatly improved under the guidance of the reviewers.

REVIEWER COMMENTS

Reviewer #1 (Remarks to the Author):

This reviewer carefully read the manuscript "Brain-wide projection reconstruction of single functionally defined neurons" by the authors. The manuscript introduces a novel method to connect in vivo neural function with morphology information (neural projections). The authors have demonstrated the scheme to label sound-evoked neurons observed by two-photon Ca²⁺ imaging by targeting techniques (single-cell plasmid electroporation and AAV) and obtain several neuronal projection data by high-resolution imaging at whole-brain scale (fMOST). This method contributes to connecting physiological function along with neural projection, and this data is the novel part of this paper. Overall, the reviewer was convinced that the manuscript would be valuable to the journal and tries to suggest some points to be improved for the publication.

We thank the reviewer very much for their support, constructive critique and helpful suggestions towards improving the quality and rigor of our manuscript. We provided detailed responses below to address the reviewer's questions point-by-point, and the revised the manuscript accordingly, as appropriate.

Major concerns:

#1. (Page 24, Line 398) The description of "0.01M PBS" is unclear. If the author used modified PBS, please describe the concentration of each of the components and pH. **We thank the reviewer for this suggestion. We used a conventional formulation of PBS that was purchased from Sigma-Aldrich. We have added a detailed description of the pH and product code in the Methods section (Page 12, line 34): "The 0.01 M PBS contained 0.138 M NaCl and 0.0027 M KCl; pH 7.40 was determined by pH meter, accurate to two decimal places (Sigma-Aldrich, cat. no. P3813-10PAK)".**

Minor concerns:

#1. (Page 1, Line 19) The section of "Main text" would be better if divided into some conventional subjects (Introduction, Results, Discussion).

We again thank the reviewer for their advice. We have divided the main text into conventional sections, including Introduction, Results, and Discussion.

Reviewer #2 (Remarks to the Author):

This is a very interesting ms, which describes a method for morphological and functional phenotyping of single neurons in vivo. The method combines 2-photon calcium imaging in awake head-fixed mice for functional definition of single cells in a region of interest (e.g. sound-evoked responses) with single cell plasmid electroporation to delineate brain-wide projections in a cell specific manner using subsequent optical tomography. The paper illustrates the method applied to the study of the auditory cortex and describes a number of observations. The approach is outstanding and it may facilitate a better morphological profiling of brain functional granularity.

We are very grateful for the reviewer's supportive comments, critical commentary and constructive suggestions that have helped us to substantially improve both the quality of our study and the rigor of our conclusions. Below, we provide point by point responses to address each of the reviewer's questions, and modified the manuscript to reflect these issues, as appropriate.

Major comments:

1- A major issue for successful dissemination of single-cell methods is feasibility. Most of the time the method works in the expert hands, but acquiring expertise may be challenging. While this is inherent to the method itself, providing a realistic view is essential. The ms will benefit from including specific assessment of the method feasibility in terms of the proportion of recovered cells from the total number of attempts. I would even consider adding a figure or a table to reinforce the critical milestones and the associated rate of success. Also, it may be important to clarify how many neurons are advisable to attempt to avoid issues with posterior morphological reconstruction in terms of separability.

We thank the reviewer for these astute and highly constructive suggestions. As suggested, we have made the following improvements:

(1) We have added a set of guidelines for troubleshooting (Table 1) that include descriptions of feasibility, such as critical steps, potential problems or confounding factors, possible explanations, and solutions for supporting the major milestones in the process.

(2) We have also added the proportion of recovered neurons and success rate of labeling to Table 2, as well as the spontaneous activities recorded before and after electroporation in Fig. 2.

(3) In our study, we found that issues with separability in posterior morphological reconstruction can be avoided by limiting each single field-of-view (200 μm \times 200 μm) to no more than four neurons in two-photon imaging. We have added this point to the revised manuscript (Page 4, line 21).

(4) We have appended several explanatory notes to help readers understand why and how each step in the protocol is important, as well as how to conduct step-by-step evaluation of the outcomes.

2- The scientific/research component of the approach should be reinforced. While authors provide examples and some quantification, the ms will benefit from reinforcing statistical report. The number of mice and experimental design should be more clearly stated in the text. Refereeing to neuron1 or 2 across figures is confusing as they seem to reflect different mice and experiments (for instance, dendritic density of neurons in fig.2d and sup.fig.5 look quite different). Questions emerge regarding any main statistical trend for neurons projecting to similar targets versus those splitting apart, or any other dimension which reduces variability. It looks like Fig.2 reflects the result of only one experiment, but supplementary figures suggest there may be some additional data. Can authors converge at some main scientific message?

We thank the reviewer for these valuable suggestions and comments. We completely agree that the paper would benefit from clarification of our experimental components and further examples. According to these suggestions, we made the following changes:

(1) We have tried to ensure that the numbers of mice and neurons are clear in all figures and in the corresponding descriptions in the manuscript. To avoid any confusion, we have added consistent labels to indicate which neurons correspond to which mice throughout all figures. For example, neuron #1 refers to the same neuron from the same mouse (mouse #1) across figures.

(2) We added descriptions of the morphology of two new example neurons that exhibited distinct functional properties. The new data are presented in Figure 4.

(3) With the two new example neurons the sample count is still too low to allow reliable interpretation of biological meanings with statistical significance. Our primary goal was to extensively test the quality of this proof-of-concept method using multiple controls and comparative experiments, i.e., high-fidelity labelling and reconstruction of long-range interhemispheric projections of accurately targeted single cells. Indeed, many more neurons were reconstructed in subsequent experiments, as listed in the newly appended data tables. However, they were largely acquired through control experiments. We would like to share our method with peers prior to addressing our own specific questions of interest. Thus, we will respectfully abstain from asserting any potentially biased or insufficiently supported scientific conclusions obtained from the limited samples that were used as demonstration in this paper.

3- The manuscript reads as a continuous piece. However, Introduction and Results will be better organized in separated in sections. Also, a more formal Discussion section may help to put the method in context, highlighting novelty and comparison with existing solutions. Importantly, the method is limited in terms of the yield. It may be argued that variability of single-cell data will prevent to make conclusions. However, there is value in adopting this approach which could eventually permit annotation of alternative high-throughput approaches based on the much fewer but morphologically and functionally identified neurons. Authors should consider these dimensions along a more detailed discussion.

We again thank the reviewer for their comments and suggestions. We have divided the main text into conventional sections (i.e., Introduction, Results, Discussion). We have also revised the Discussion section to incorporate more detail, compare our method with other existing approaches, and highlight the novelty of our method. In particular, we emphasize that this high-precision method can be broadly adopted by the research community to complement and further examine data generated by existing high-throughput methods.

Other comments

4- Some aspects of the method should be better explained. For instance, electroporation is a critical state. Are authors monitoring the cell response? How do they know a cell is successful electroporated? Is the intracellular dye used for this purpose? How much is the pipette advanced towards the soma? Is there any potential issue with viral spilling over from

the pipette? How many cells remain functional after electroporation and can be tracked across days?

We thank the reviewer for raising these questions. We provide point-by-point replies to each question below:

- (1) We have appended new data (Fig. 2) that show spontaneous activity observed by monitoring for several minutes before and after electroporation;**
- (2) Successful electroporation was verified by immediate filling of the neuron body with the extra OGB-1-K⁺ dye, as now described in main text;**
- (3) After the tip of the pipette was in visual contact with the edge of cell body, it was further advanced very gently to the center of cell body, with a mean distance of 2 μm , as now described in the extended methods section;**
- (5) Although there viral spill-over from the pipette is a valid concern, we found no effects from this issue in our results. This is could be explained by a key point in our method, i.e., the combination of single-cell Cre plasmid electroporation and nearby local AAV injection to utilize Cre to unlock and amplify robust fluorescence expression in targeted neurons. Using this strategy, only the electroporated neurons express robust fluorescence signals. We have explained this point in more explicit detail in the revised manuscript.**
- (6) We recorded spontaneous activity for several minutes after electroporation to confirm that the electroporated neuron remained alive. In our experiment, approximately 92% of the neurons remained functional after electroporation (Table 2 and Fig. 2), and about 82% neurons could be tracked across days after electroporation (Table 2).**

5- Some statements should be clarified: line 73 “complete and robust?” ; line 116 “bilateral hemispheres?” ; line 143 “confirmed interhemispheric cortico-cortical projections by single-cell electroporation..?” If I understand correctly interhemispheric cortico-cortical projections are determined by AAV2Retro virus followed by single-cell electroporation of the Cre-GFP plasmid

We thank the reviewer for bringing these unclear points to our attention. We have revised our statements to improve their clarity (see below).

- (1) We have changed “complete and robust” to “complete” (Page 3, line 12);**
- (2) We have changed “neuron 1 projected extensively to bilateral hemispheres” to “neuron #1 exhibited a bilateral projection pattern” (Page 5, line 13);**
- (3) Yes, the understanding is correct. AAV2/2-Retro virus injection was used in a set of control experiments to verify that our 2-SPARSE method could reliably label long-range interhemispheric projections (now with a new subtitle “Evaluation of the quality of 2-SPARSE**

for long-range projection labelling”). We again thank the reviewer for helping us to ensure that our experiments are clear and easily understood. We have revised the text to avoid further potential misunderstandings (Page 6, line 17-25).

6- Fig.2: please, clarify what are abbreviations used in panel b. Regarding this, it looks like neurons have their soma in TEa not in AUD, so strictly speaking they are neurons from the temporal cortex. Is that correct?

Thank you for this comment.

(1) We have clarified the abbreviations used in Fig. 3b (old Fig. 2b) in its accompanying figure legend (see below).

“AUD, auditory areas; TEa, temporal association areas; SSs, supplemental somatosensory area; LA, lateral amygdalar nucleus; CP, caudatoputamen; c, contralateral; i, ipsilateral.”

(2) No, the neurons have their somata in AUD not in TEa. To avoid misunderstanding, we have changed the 3D viewing angle for brain areas in Fig. 3c (old Fig. 2c). The figure now clearly shows that both neurons have their somata in AUD.

7- Fig.2d and e, please use the same color code. For fig.2d, how many planes are stacked per each cell? At L1 to L4 limits comparable for the two neurons shown here? Please, add details in caption.

We thank the reviewer for this highly constructive idea and their suggestions.

(1) We have changed the color coding of Fig. 3d and e (old Fig.2d and e) so that colors are consistent across figures.

(2) For Fig. 3d (old Fig. 2d), 300 imaging planes (600 μm) were stacked per each cell to reveal a dendritic tree.

(3) We have added more details relevant to understanding Fig. 3 (old Fig. 2) in its accompanying legend.

(4) For these two example neurons the depth range limit from L1 to L4 was sufficient to fully encompass their complete dendritic trees.

Reviewer #3 (Remarks to the Author):

In this manuscript, Brain-wide projection reconstruction of single, functionally defined neurons the authors attempt to tackle an important challenge in behavioral neuroscience, obtaining the combined functional characterization and neuronal morphology of individual neurons. As the authors point out, this is a technically challenging problem, especially doing this systematically or at scale. However, this manuscript does not provide any novel

technological enhancements that would make such combined analysis possible routinely in a brain area of interest. Further, there are no new biological insights or conclusions from the few neurons characterized in this study. For these reasons this paper should not be published in its current form in Nature Communications.

We thank the reviewer for the critical commentary. We are also grateful for the opportunity to better explain the novelty and significance of our innovation to the research community. In the substantially revised version of this manuscript, the method itself is explained in further detail to facilitate comparison with other, existing methods to better highlight its unique advantages. In particular, our method successfully pushes the horizon of conventional single-cell electrophysiology-filling methods (both biocytin-based and transfection-based), which currently cannot reliably label long-range interhemispheric projecting axons.

Regarding the reviewer's concerns and suggestions regarding the feasibility of routinely performing our method, we performed experiments with consecutive success and also extended into deeper cortical areas.

Moreover, while we respect the reviewer's assertion that no new biological insights were presented in this paper, our primary goal in this proof-of-concept study was to extensively test the quality of the 2-SPARSE method through control and comparative experiments. We would like to share our method with peers prior to addressing our own specific questions of interest, and thus we abstained from asserting any scientific conclusions obtained from the limited samples used as demonstration in this paper. The utility and versatility of this process should be obviously apparent to any workers in the field, and we are extremely confident that adoption of this method as a complement to other high-throughput methods will enable exploration of several long-standing questions that could not be previously undertaken. We again thank the reviewer for taking the time to share their opinions.

Specific Comments:

1) Electrophysiological recordings combined with juxtacellular labeling and morphological reconstructions are not new. Multiple published works exist, some nice examples can be found in the following (Deschênes et al., 1998; Furuta et al., 2011; Li et al., 2017). The authors do not reference any earlier published work that do this. Are the authors suggesting that their 2P calcium imaging combined with targeted labeling is a higher-throughput approach? If so, this is not borne out by the low yield of reconstructed neurons in each brain or the total neurons characterized in this manner in this study.

Deschênes, M., Veinante, P., and Zhang, Z.-W. (1998). The organization of corticothalamic projections: reciprocity versus parity. *Brain Research Reviews* 28, 286–308.

Furuta, T., Deschênes, M., and Kaneko, T. (2011). Anisotropic distribution of thalamocortical boutons in barrels. *J Neurosci* 31, 6432–6439.

Li, L., Ouellette, B., Stoy, W.A., Garren, E.J., Daigle, T.L., Forest, C.R., Koch, C., and Zeng, H. (2017). A robot for high yield electrophysiology and morphology of single neurons in vivo. *Nature Communications* 8, 15604.

We appreciate the reviewer's mention of classical biocytin-based labeling methods, as well as modern transfection-based methods. As discussed above, our new method pushes the horizon set by these existing methods. We have added these suggested references to the revised manuscript and discussed them in greater detail in the Discussion section. It should be noted that we made no assertions suggesting that our method was high-throughput. In fact, to the contrary, our approach emphasizes detailed, high-precision investigation, which can serve as a powerful complement to other high-throughput methods. In the modified Discussion section we suggest that our method is uniquely advantageous for targeting sparse and widely distributed neuronal subpopulations with defined functional features. In investigations of this nature, high-throughput methods that have inherently high noise but lack single-cell specificity can yield highly ambiguous results.

2) There are no new biological results that emerge that relate neuronal function and their structure. Are there neurons with similar functional properties that are structurally very different or alternatively, are there neurons with unique structural signatures that have distinct functional correlates.

We thank the reviewer for this highly constructive comment. Indeed, we found that there are neurons with similar functional properties but which are structurally very different (shown in Fig. 3). We have added examples of neurons with unique structural signatures but distinctly different functional correlates (Fig. 4). However, these are just examples and we respectfully abstain from including conjecture, potentially biased interpretations, or insufficiently supported scientific conclusions based our limited sample set.

3) There are two neurons shown in figure 2 that have the same tonal properties but with different morphologies, but the data is too anecdotal to draw any conclusions. More exemplars of the two morphologies would be needed.

We thank the reviewer for this suggestion, and we agree that more examples would support our results. In the revised manuscript, we sampled many more neurons and presented the results in detail (new Fig. 4, Supplementary Fig. 8, Table 3).

4) The data is not convincing enough to ensure completeness of cell filling and reconstructions. For instance, is the result of the two different morphologies in figure 2 real, is the non-contralateral projecting neuron just an incomplete fill and reconstruction? A retrograde tracing experiment that is contingent on targeting both the contralateral AUDp and CP should be done to corroborate this result.

We thank the reviewer for their advice.

As suggested, we performed a retrograde tracing experiment from the contralateral AUDp (Figure 5). As the distance of the contralateral cortico-cortical projection was 9.6 mm long, more than twice that of the 4.7 mm ipsilateral cortico-striatal projection, we believe that the cell filling and reconstruction experiments in the contralateral-projecting neurons are sufficient to ensure completeness.

In addition, we performed morphological reconstructions of neurons in the AUD using conventional AAV-based labelling for comparison with our 2-SPARSE approach. The results of conventional AAV-based labelling show clear agreement with the results of 2-SPARSE (Fig. 6a).

Furthermore, we used our approach to reconstruct the morphology of IT neurons in the motor cortex (Fig. 7), which confirmed that our method could indeed be generalized for application in more complex IT neurons (>21 cm of axonal length).

These three experiments each support the completeness of cell filling and reconstruction using the 2-SPARSE method.

5) Providing some context for the completeness of reconstructions would be important. For instance, from comparisons to the mesoscale data from the Allen institute what is the expectation for brain areas targeted by layer 2/3 neurons from AUDp. What are expectations for number of morphological types?

Following the reviewer's advice, we compared our single-cell data with mesoscale data from the Allen institute to examine the completeness of reconstructions performed in this manuscript. However, it should be noted that population-level statistical completeness of projection patterns does not indicate the same degree of completeness per each single-cell. To address your question, we performed a cell-by-cell validation in a series of control experiments using AAV2/2-Retro to establish the baseline presence of a long-range interhemispheric projection.

In our work, we simply classified the observed morphologies into two categories: those with and those without interhemispheric long-range projections.

6) Additional proof for completeness of cell fills by this approach could also come from indirect experiments wherein morphological reconstructions are done in AUDp cells labeled using established viral reagents showing concordance between the two methods.

We are very glad for the reviewer's advice. As suggested, we conducted this experiment (new Fig. 6) and found no significant differences in measurements of dendritic and axonal lengths between methods, nor in the total numbers of axonal and dendritic branches between the neurons labelled by 2-SPARSE and those labelled using viral reagents (Fig. 6c).

7) It is not clear that the approach used in this study will generalize to other neuron types including IT neurons from motor cortex for instance that appear to be far more complex (>35cms of axonal length) than any neurons reconstructed in this study.

We appreciate this suggestion and agree that it is highly germane to demonstrating the versatility of our method. We have added a reconstruction of IT neurons, as recommended (new Fig. 7).

8) Can deeper layer cells be targeted efficiently?

We agree that this is also a highly valuable experiment for illustrating the targeting capabilities of this method and we have subsequently performed the requested experiment in layer 5 of the motor cortex (new Fig. 7).

We again thank the reviewer for their gracious time and consideration in helping to ensure the rigor and quality of our conclusions. This process has greatly improved our paper through the invaluable guidance and relevant suggestions of the reviewer, and has taught us much that will inform our future investigations and writing.

REVIEWERS' COMMENTS

Reviewer #2 (Remarks to the Author):

The authors have performed a thoroughful revision of the ms. They provide new data of single cells filled from different regions (motor cortex, deep layers) and new cells with different tone-tuning responses. In addition, they provide a new figure reporting new experiments to compare their single-cell method with conventional AAV sparse labeling approach. The revised ms includes a more clear description of the relevant steps and troubleshooting to address my concerns regarding feasibility.

One of my major comments was to reinforce the experimental component by reporting data leading to new knowledge. The authors have taken actions to ensure reporting the number of mice used and adding more data, but chose not to reach conclusions. They seem to prefer to share the method by illustrating examples rather than making conclusions. While that would not be my personal take, I feel the authors should make their own choice. The overall impression of the paper is now more robust with the new data mentioned above. Therefore I found this version improved and all my concerns addressed.

I congratulate the authors for a serious revision and a nice work. I don't see any additional issue.

The editor asked me for my view on whether the authors have addressed Reviewer 3's concerns:

In my opinion R3's concerns are based on biocytin-based labeling methods, but the authors here report a transfection-based method instead. They use biocytin filling as a control. With biocytin filling you may have issues with diffusion and incomplete labeling. Transfection methods, in which a single cell incorporates the reporter transcript overcome this limitation and because of expression extends to the whole cell morphology coverage is perfect. I feel this reviewer missed the opportunity to see the novelty of the method and this possibly influences his/her technical criticisms. From R3's suggested references only the Nat Comm paper describes a robot carrying electroporation of DNA plasmid for high yield recording and labeling. In contrast, this method is meant to be high-precision allowing for targeting sparse neurons based in ephys features. So, the answer to your first two questions is yes, I disagree with his/her overall assessment and no, R3 comments did not change my recommendation about this paper. I strongly endorse publication.

Regarding R3's points 4-6. Here again, I feel R3 is having biocytin filling in mind and not the authors' approach. Their answer is clear: a 9.6 mm long recovery sounds pretty good for a contralateral cell. They compare their method with AAV-based labeling and different neuronal types, including a long motor cortex neuron with >20cm long axonal length. Moreover, in response to points 5 and 6 they compared data with Allen BA information and performed new validation experiments including in Fig.6. To me there is no doubt concerns are addressed.

We would like to express our deep appreciation again to the reviewers for the constructive comments and suggestions towards improving our manuscript (NCOMMS-21-07308A).

REVIEWER COMMENTS

Reviewer #2 (Remarks to the Author):

The authors have performed a thoroughful revision of the ms. They provide new data of single cells filled from different regions (motor cortex, deep layers) and new cells with different tone-tuning responses. In addition, they provide a new figure reporting new experiments to compare their single-cell method with conventional AAV sparse labeling approach. The revised ms includes a more clear description of the relevant steps and troubleshooting to address my concerns regarding feasibility.

One of my major comments was to reinforce the experimental component by reporting data leading to new knowledge. The authors have taken actions to ensure reporting the number of mice used and adding more data, but chose not to reach conclusions. They seem to prefer to share the method by illustrating examples rather than making conclusions. While that would not be my personal take, I feel the authors should make their own choice. The overall impression of the paper is now more robust with the new data mentioned above. Therefore I found this version improved and all my concerns addressed.

I congratulate the authors for a serious revision and a nice work. I don't see any additional issue.

The editor asked me for my view on whether the authors have addressed Reviewer 3's concerns:

In my opinion R3's concerns are based on biocytin-based labeling methods, but the authors here report a transfection-based method instead. They use biocytin filling as a control. With biocytin filling you may have issues with diffusion and incomplete labeling. Transfection methods, in which a single cell incorporates the reporter transcript overcome this limitation and because of expression extends to the whole cell morphology coverage is perfect. I feel

this reviewer missed the opportunity to see the novelty of the method and this possibly influences his/her technical criticisms. From R3's suggested references only the Nat Comm paper describes a robot carrying electroporation of DNA plasmid for high yield recording and labeling. In contrast, this method is meant to be high-precision allowing for targeting sparse neurons based in ephys features. So, the answer to your first two questions is yes, I disagree with his/her overall assessment and no, R3 comments did not change my recommendation about this paper. I strongly endorse publication.

Regarding R3's points 4-6. Here again, I feel R3 is having biocytin filling in mind and not the authors' approach. Their answer is clear: a 9.6 mm long recovery sounds pretty good for a contralateral cell. They compare their method with AAV-based labeling and different neuronal types, including a long motor cortex neuron with >20cm long axonal length. Moreover, in response to points 5 and 6 they compared data with Allen BA information and performed new validation experiments including in Fig.6. To me there is no doubt concerns are addressed.

Reply: Since there is no additional issue to be addressed, we thank the reviewer again for helping to improve our manuscript.